# Scalable Universal T-Cell Receptor Embeddings from Adaptive Immune Repertoires

**Paidamoyo Chapfuwa**[1*], **Ilker Demirel**[2†], **Lorenzo Pisani**[1]
**Javier Zazo**[3], **Elon Portugaly**[3], **H. Jabran Zahid**[1], **Julia Greissl**[1]
[1]Microsoft Research, Redmond, USA; [2]MIT, USA; [3]Microsoft Research, Cambridge, UK
{pchapfuwa,lopisani,javierzazo,elonp,hzahid,jugreiss}@microsoft.com
demirel@mit.edu

## Abstract

T cells are a key component of the adaptive immune system, targeting infections, cancers, and allergens with specificity encoded by their T cell receptors (TCRs), and retaining a memory of their targets. High-throughput TCR repertoire sequencing captures a cross-section of TCRs that encode the immune history of any subject, though the data are heterogeneous, high dimensional, sparse, and mostly unlabeled. Sets of TCRs responding to the same antigen, *i.e.*, a protein fragment, co-occur in subjects sharing immune genetics and exposure history. Here, we leverage TCR co-occurrence across a large set of TCR repertoires and employ the GloVe (Pennington et al., 2014) algorithm to derive low-dimensional, dense vector representations (embeddings) of TCRs. We then aggregate these TCR embeddings to generate subject-level embeddings based on observed *subject-specific* TCR subsets. Further, we leverage random projection theory to improve GloVe's computational efficiency in terms of memory usage and training time. Extensive experimental results show that TCR embeddings targeting the same pathogen have high cosine similarity, and subject-level embeddings encode both immune genetics and pathogenic exposure history.

## 1 Introduction

Low-dimensional representations that capture meaningful qualities of the input data they compress are one of the foundations of modern machine learning. In text and imaging modalities, these embeddings have reached a high level of sophistication and have significantly advanced our ability to build tools such as automatic image captioning or visual questions answering systems (Dosovitskiy et al., 2021; Vaswani et al., 2017). Recently, significant progress has been made in deriving embeddings in life science domains as diverse as proteins (Lin et al., 2023) and the RNA expression levels of cells (Rosen et al., 2023). However, modalities in the life sciences bring their own set of challenges due to a lack of data at scale, data heterogeneity, and data structures that violate assumptions made in text and imaging domains. These challenges require novel approaches to build high-quality representations. Here, we focus on learning representations of T-cell receptors (TCRs) and TCR repertoires.

The primary function of T cells is to identify fragments of foreign proteins, known as *antigens*, derived from viruses, bacteria, and cancerous cells, and to kill the cells in which these proteins are found. These antigens are presented by human leukocyte antigens (HLAs) (Zinkernagel & Doherty, 1974). Each subject has many HLAs, and HLA diversity is high in the human population (Hughes & Yeager, 1998). HLA frequencies are *power-law distributed*, so subjects typically share some but not all of their HLAs. T cells are specific to both antigens and the presenting HLA, with specificity encoded by the TCR (Davis & Bjorkman, 1988). To effectively clear infections, T cells clonally expand ($>1000-$fold) once they encounter a cognate antigen (Burnet, 1976), which significantly increases their likelihood of being sampled and sequenced. As a consequence, subjects sharing HLAs and pathogenic exposure have a significantly higher likelihood of sharing a subset of TCRs

---

*To whom correspondence should be addressed
†Work done principally during an internship at Microsoft Research

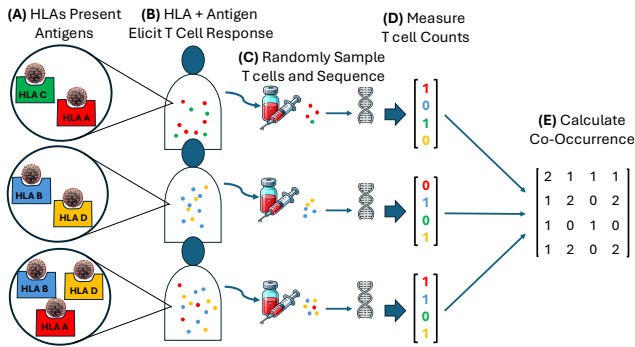

Figure 1: A schematic of the data generation process. Each subject has a set of HLA types (A) which together with exposure history define their TCR repertoire (B). A subset of these TCRs ($O(10^6)$) are sequenced (C). Using a large number of repertoires, we generate TCR co-occurrence matrix that is the input to our model (D, E).

than would be expected randomly. Thus, sets of TCRs specific to a particular HLA-pathogenic exposure combination co-occur in subsets of subjects who share the HLA and pathogenic exposure. It is possible to identify millions HLA-associated TCRs, and previous work has demonstrated that co-clustering yields subsets of TCRs that map to specific pathogens (Zahid et al., 2024; May et al., 2024). Previous work applying deep learning to TCRs has focused on *supervised* learning of TCR protein features for antigen-binding prediction and repertoire classification (Zhang et al., 2021; Zhao et al., 2023; Widrich et al., 2020; Sidhom et al., 2021).

Here, we infer TCR embeddings leveraging co-occurrence patterns, which encodes exposure to hundreds of common pathogens and HLA-associations (DeWitt III et al., 2018). We then aggregate all TCRs observed in a subject's repertoire to derive a subject-level embedded representation that encodes both the subject's HLAs and their immune history. See Figure 1 for an overview of the data generation process. TCR repertoires are uniquely positioned to capture which TCRs share an immunological context because they probe deeply into an individual's immune system (Robins, 2013). Moreover, TCR embeddings represent TCRs within an immunological context, such as being part of a group of TCRs responding to the SARS-CoV-2 spike protein or a group of Cytomegalovirus (CMV) TCRs. When aggregated into repertoire embeddings, this represents a low dimensional snapshot of an individual's immune history.

The key contributions of this paper are as follows:

- We show that TCR embeddings can be learned from co-occurrences using the GloVe (Global Vectors for Word Representation) algorithm (Pennington et al., 2014).
- We provide a proof-of-concept demonstrating that a simple aggregation of TCR embeddings into *repertoire* embeddings captures both the immune genetics and pathogenic exposure history of individuals.
- We demonstrate TCR embeddings get richer when more TCRs and repertoires are used.
- We leverage random projection theory to improve GloVe's computational complexity in terms of memory usage and training time.
- We explore the topology of the embedding space, finding that vectors of TCRs associated with the same antigen and repertoires with similar genetic or exposure profiles have higher cosine similarities.

## 2 LEARNING TCR EMBEDDINGS

Co-occurrence of TCRs across repertoires has been employed to discover TCRs targeting the same exposures (Emerson et al., 2017; DeWitt III et al., 2018) and HLAs (Zahid et al., 2024). A similar phenomenon has been observed in natural language processing, where words with similar meanings co-occur in similar contexts. The GloVe (Pennington et al., 2014) algorithm exploits this property to learn semantically meaningful word embeddings from co-occurrences. Here, we use TCR$\beta$ se-

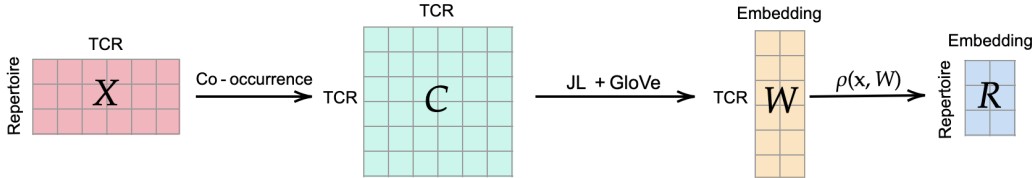

Figure 2: Overview of the modeling approach. We compute a TCR-TCR co-occurrence count matrix $C \in \mathbb{Z}^{K \times K}$ from a repertoire-TCR binary matrix $X \in \{0,1\}^{N \times K}$, namely $C = X^T X$. Leveraging the GloVe (Pennington et al., 2014) algorithm and the Johnson-Lindenstrauss (JL) transform (Lindenstrauss & Johnson, 1984), we infer low-dimensional TCR embeddings $W \in \mathbb{R}^{K \times d}$ from TCR co-occurrence $C$. We generate repertoire embeddings $R^{N \times d}$ via a simple pooling function $\rho(\boldsymbol{x}, W)$ in Equation (8).

quences defined by their V gene, J gene, and CDR3. The proposed method is general and could also be applied to TCR$\alpha$-only sequencing or paired TCR$\alpha$+TCR$\beta$ sequencing.

We apply a modified version of the GloVe algorithm to learn immunologically meaningful TCR embeddings. Rather than using a random weight initialization scheme, we leverage random projection theory to initialize GloVe, leading to improved memory usage and training time while enabling training with only a fraction of the data. We calculate TCR co-occurrence statistics from TCR repertoire measurements of $N$ subjects across $K$ TCRs. Let $\mathcal{T} = \{t_1, t_2, \ldots, t_K\}$ be the set of $K$ TCRs we consider, and $\boldsymbol{x}_n \in \{0,1\}^K$ the $n$-th TCR repertoire s.t. $\boldsymbol{x}_n(k) = 1$ means TCR $t_k$ is present in an individual and absent otherwise. We use a binary indicator for TCR presence/absence instead of clonal frequencies for more robust learning. We use TCR repertoire measurements of $N$ individuals and denote by $X \in \{0,1\}^{N \times K}$ the sparse and binary repertoire-TCR matrix where $K >> N$. Figure 2 illustrates our end-to-end modeling approach for $(i)$ learning low-dimensional TCR embeddings from co-occurrence and $(ii)$ generating repertoire embeddings via a simple pooling function $\rho(\cdot)$ in Equation (8).

## 2.1 USING THE GLOVE ALGORITHM TO LEARN FROM TCR CO-OCCURRENCES

The GloVe algorithm uses co-occurrence statistics across documents combining global matrix factorization methods (*e.g.*, latent semantic analysis (Deerwester et al., 1990)) and local context window methods (*e.g.*, Word2Vec (Mikolov et al., 2013a)). We adopt the algorithm for TCRs. Let $C$ denote the TCR-TCR co-occurrence matrix where $C(i,j)$ is the number of repertoires that contain both TCRs $i$ and $j$. While constructing $C$ for words requires a rather involved scan over the documents, in our case it is a simple matrix product $C = X^\mathsf{T} X$. We exploit this difference to develop a faster algorithm with lower training time and space complexity. Because $X$ is a sparse binary matrix, we leverage distributed techniques for matrix multiplication using only the *non-zero elements* in $X$ in Spark (Zaharia et al., 2010). Refer to Appendix A.4 for details on the Spark algorithm used to compute $C$.

We learn TCR embeddings by minimizing the following GloVe loss function $\mathcal{L}(\bar{W}, \bar{\boldsymbol{b}}, \tilde{W}, \tilde{\boldsymbol{b}}; C)$ formulated as:

$$\mathcal{L}(\bar{W}, \bar{\boldsymbol{b}}, \tilde{W}, \tilde{\boldsymbol{b}}; C) = \sum_{i,j} f(C_{ij}) \big(\langle \bar{\boldsymbol{w}}_i, \tilde{\boldsymbol{w}}_j \rangle + \bar{b}_i + \tilde{b}_j - \log C_{ij}\big)^2, \tag{1}$$

where $\{\bar{\boldsymbol{b}}, \tilde{\boldsymbol{b}}\} \in \mathbb{R}^K$ are bias terms of the TCR embeddings $\{\bar{W}, \tilde{W}\} \in \mathbb{R}^{K \times d}$ that allow for more flexible modeling of the *marginal* frequency of occurrence. Previous studies have shown that bias terms are correlated to the marginal frequency occurrence (Shazeer et al., 2016), which is consistent with our results. In practice, for a given TCR the final embedding $\boldsymbol{w}$ is the average of the learned embeddings $\bar{\boldsymbol{w}}$ and $\tilde{\boldsymbol{w}}$. The weighting function $f : \mathbb{Z}_+ \to \mathbb{R}_+$ prevents commonly occurring TCRs from dominating the loss. We deviate from GloVe's weighting function by specifying a monotonic transformation of the co-occurrence counts $f(\cdot) = s_0 \mathbb{1}(C_{ij} > 0) + s C_{ij}^\alpha$ according to Shazeer et al. (2016), where hyperparameters are set to $\{s_0 = 0.1, s = 0.25, \alpha = 0.5\}$ and $\mathbb{1}(a)$ is an indicator function s.t. $\mathbb{1}(a) = 1$ if $a$ holds or $\mathbb{1}(a) = 0$ otherwise. This approach yields good performance without hyper-parameter tuning. Similar to GloVe, we only train on the *non-zero elements* of the TCR-TCR co-occurrence matrix, which is computationally efficient given that $C$ is very sparse

(approximately 80% of the entries in $C$ are zero in our case). Note that GloVe optimizes with respect to the log-co-occurrences to preserve the ratios of co-occurrence probabilities, which can also be considered as a simple monotonically increasing transform for regularizing the effect of large co-occurrences. Model parameters $\{\bar{W}, \bar{\boldsymbol{b}}, \tilde{W}, \tilde{\boldsymbol{b}}\}$ of the cost function in Equation (1) are optimized using stochastic gradient descent on minibatches from $C$.

## 2.2 SCALABLE LEARNING VIA JOHNSON-LINDENSTRAUSS TRANSFORM

**GloVe does not scale well.** To minimize Equation (1), we first compute the co-occurrence matrix $C$, which requires calculating $K^2$ dot-products of $N$-element vectors, resulting in $O(NK^2)$ time complexity which scales quadratically with the number of TCRs. This calculation becomes very expensive when learning embeddings for millions of TCRs. Additionally, we need to compute and sum $O(K^2)$ terms (depending on the sparsity of $C$) to backpropagate the loss in Equation (1) in a single training epoch, which adds to the computational complexity of GloVe. To address these issues, we develop a learning scheme where we *initialize* GloVe embeddings with carefully constructed initial TCR embeddings and improve GloVe's computational complexity in training time and memory usage.

**Fast random projections can reduce dimensionality while preserving structure.** We reduce the dimensionality of $C$ using random projections (Achlioptas, 2003; Roughgarden & Valiant, 2024). Precisely, let $P$ be a $K \times d$ random matrix where $d \in \mathbb{Z}_+$ is the desired dimensionality of the TCR embeddings. We compute

$$W^{\mathrm{JL}} = \frac{1}{\sqrt{d}} \underbrace{X^{\mathsf{T}}(X}_{=C} P) \,, \tag{2}$$

where $W^{\mathrm{JL}}$ is $K \times d$ and we denote by $\boldsymbol{w}_i^{\mathrm{JL}}$ the $i$-th row of $W^{\mathrm{JL}}$ for $i \in \{1, 2, \ldots, K\}$. Here JL stands for Johnson-Lindenstrauss transform (see Lindenstrauss & Johnson (1984)). $XP$ requires $d \times N$ dot-products of $K$-element vectors, and $X^{\mathsf{T}}(XP)$ requires $d \times K$ dot-products of $N$-element vectors, resulting in $O(dNK)$ time-complexity. Furthermore, when $P$ is sampled carefully according to Theorem 2.1, the transform given by Equation (2) approximately preserves the pairwise L2-distances between the rows of $C$.

**Theorem 2.1** (Theorem 1.1 in Achlioptas (2003)). *Let the elements of $P$ be i.i.d. drawn as follows:*

$$P(a, b) = \sqrt{3} \times \begin{cases} -1, & \text{with probability } 1/6 \\ 0, & \text{with probability } 2/3 \\ 1, & \text{with probability } 1/6 \,. \end{cases} \tag{3}$$

*Given $\epsilon, \beta > 0$, if $d \geq \frac{4+2\beta}{\epsilon^2/2 - \epsilon^3/3} \log K$, we have, for all $i, j \in \{1, 2, \ldots, K\}^2$,*

$$(1 - \epsilon) \left\| \boldsymbol{c}_i - \boldsymbol{c}_j \right\|^2 \leq \left\| \boldsymbol{w}_i^{\mathrm{JL}} - \boldsymbol{w}_j^{\mathrm{JL}} \right\|^2 \leq (1 + \epsilon) \left\| \boldsymbol{c}_i - \boldsymbol{c}_j \right\|^2 \,, \tag{4}$$

*with probability at least $1 - K^{-\beta}$.*

This means that if the L2-distances between $\boldsymbol{c}_i \in \mathbb{R}^K$ reliably correlate with the co-occurrence patterns of the TCRs, one can use lower dimensional $\boldsymbol{w}_i^{\mathrm{JL}} \in \mathbb{R}^d$ as TCR embeddings instead, where $d = O(\epsilon^{-2} \log K)$ for any distortion $0 < \epsilon < 1$ (Dasgupta & Gupta, 2003; Lindenstrauss & Johnson, 1984).

## 2.3 JL-GLOVE: IMPROVING GLOVE COMPLEXITY WITH JL INITIALIZATION

While the JL transformation has lower complexity than GloVe, *i.e.*, $O(dNK)$ *vs.* $O(NK^2)$, it has 2 key limitations: ($i$) JL embeddings are of lower quality than GloVe's as shown in Figure 4, and thus they do not perform as well on downstream tasks, as discussed in Appendix A.11; ($ii$) L2-distances between co-occurrence columns $\boldsymbol{c}_i$ are preserved when $d = O(\epsilon^{-2} \log K)$, which significantly increases the TCR embedding dimensions as we scale $K$ to millions of TCRs. Therefore, we initialize GloVe embeddings in Equation (1) with JL embeddings $\tilde{W}^{\mathrm{JL}}$ in Equation (6) resulting in faster convergence and allowing us to achieve good performance using only a fraction of the co-occurrence matrix $C$, as shown in Figure 3. Note that if TCRs $t_i$ and $t_j$ have similar co-occurrence patterns,

*i.e.*, small $\|\boldsymbol{c}_i - \boldsymbol{c}_j\|$, we expect their respective JL embeddings $\left\|\boldsymbol{w}_i^{\text{JL}} - \boldsymbol{w}_j^{\text{JL}}\right\|$ L2-distances, to be small. In contrast, the GloVe objective in Equation (1), optimizes for cosine similarity, *i.e.*, TCRs $t_i$ and $t_j$ will have high cosine similarity $\langle \boldsymbol{w}_i, \boldsymbol{w}_j \rangle = \|\boldsymbol{w}_i\| \, \|\boldsymbol{w}_j\| \cos(\boldsymbol{w}_i, \boldsymbol{w}_j)$. To bridge this gap, we leverage the geometric relationship

$$\|\boldsymbol{w}_i - \boldsymbol{w}_j\|^2 = \|\boldsymbol{w}_i\|^2 + \|\boldsymbol{w}_j\|^2 - 2\langle \boldsymbol{w}_i, \boldsymbol{w}_j \rangle. \tag{5}$$

Given Equation (5), it is evident that *unit-normalizing* JL embeddings $W^{\text{JL}}$ enables alignment with the GloVe objective function in Equation (1), thus resulting in faster convergence, *i.e.*, reduced training time. Moreover, a normalized JL transform also accounts for varying *marginal occurrences of TCRs*. Refer to the Appendix A.2 for a more in-depth discussion on this aspect. Henceforth, we refer to the proposed GloVe algorithm initialized with JL as JL-GLOVE.

**Normalizing JL Transform** We aim to normalize $W^{\text{JL}}$ without violating the JL transform properties in Theorem 2.1 or computing $C$. Naively, one could compute $C$, unit-normalize it to get $\tilde{C}$, s.t. $\tilde{\boldsymbol{c}}_i = \frac{\boldsymbol{c}_i}{\|\boldsymbol{c}_i\|}$, see illustration in Equation (12). After this transformation, the comparison of pairwise distances $\|\tilde{\boldsymbol{c}}_i - \tilde{\boldsymbol{c}}_j\|$ will not be affected by *marginal occurrences*. Importantly, $\|\tilde{\boldsymbol{c}}_i - \tilde{\boldsymbol{c}}_j\|$ should be smaller for co-occurring TCRs and thus preserved after the JL-transform

$$\tilde{W}^{\text{JL}} = \frac{1}{\sqrt{d}} \tilde{C} P. \tag{6}$$

Hence, rows of $\tilde{W}^{\text{JL}}$ in Equation (6), $\tilde{\boldsymbol{w}}_i^{\text{JL}} \in \mathbb{R}^d$ could be used to initialize GloVe TCR embeddings. However, to compute $\tilde{C}$, one needs $C = X^\mathsf{T} X$ first, which is quadratic-time in the number of TCRs $K$. We instead avoid constructing $C$ and approximate $\tilde{W}^{\text{JL}}$ in linear-time:

**Proposition 2.2.** *$\tilde{W}^{\text{JL}}$ in Equation (6) can be approximated in $O(dNK)$ time when $P$ is constructed as in Theorem 2.1, by computing $W^{\text{JL}}$ in Equation (2) first and normalizing its rows. See proof in Appendix A.1.*

The JL transform has been leveraged in various problems in the literature for its ability to reduce computational complexity (Bingham & Mannila, 2001; Akselrod-Ballin et al., 2011; Argerich et al., 2016; Canzar et al., 2024). However, its normalized version, which can be approximately computed in linear-time as we argue in Proposition 2.2, is, to the best of our knowledge, a novel contribution. The proposed JL-GLOVE algorithm is general and can be re-purposed to other problems with high-dimensional and sparse counts data, *e.g.*, B-cell receptor repertoires (Chen et al., 2022). Additionally, constructing $P$ according to Theorem 2.1, further reduces the computational complexity due to the sparsity of $P$. Refer to Appendix A.5 for details on sparse matrix algorithm used to compute $\tilde{W}^{\text{JL}}$.

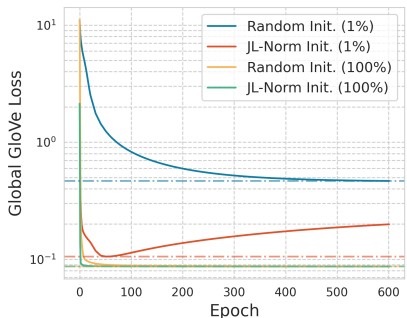

Figure 3: Comparison of GloVe loss in Equation (1) over $C$, when TCR embeddings are initialized to random *vs.* JL-Norm ($\tilde{W}^{\text{JL}}$), and when 100% *vs.* 1% of $C$ is used ($K = 65,751$; $d = 100$).

**Comparisons of GloVe initialization schemes** We initialize GloVe embeddings in Equation (1) with *unit-normalized* JL embeddings $\tilde{W}^{\text{JL}}$ in Equation (6). Figure 3 demonstrates that initializing GloVe embeddings with $\tilde{W}^{\text{JL}}$ (JL-Norm) and training on 1% of the co-occurrence data $C$ results in a loss that approaches the minimum loss achieved when using the entire dataset (100% of $C$). Interestingly, the loss starts to increase if one keeps training, since the embeddings start over-fitting to the 1% of the co-occurrences. We address this using a separate validation subset of co-occurrences for early stopping and use another 1% of $C$ to verify that the corresponding validation loss is almost a perfect proxy for the global GloVe loss which is calculated for the entire $C$ matrix. Note that *random initialization* of GloVe using a subset of the co-occurrence data (1%) can never attain the minimum loss of JL-Norm initialization (1%), thus demonstrating that $\tilde{W}^{\text{JL}}$ contains co-occurrence information beyond what is contained in the 1% of $C$ used in training. Moreover, JL-Norm initialization converges faster than random initialization even if the entire matrix $C$ is feasible to compute and is used for training. This faster convergence becomes critical as the number of TCRs increases. We provide additional comparisons in Appendix A.11.

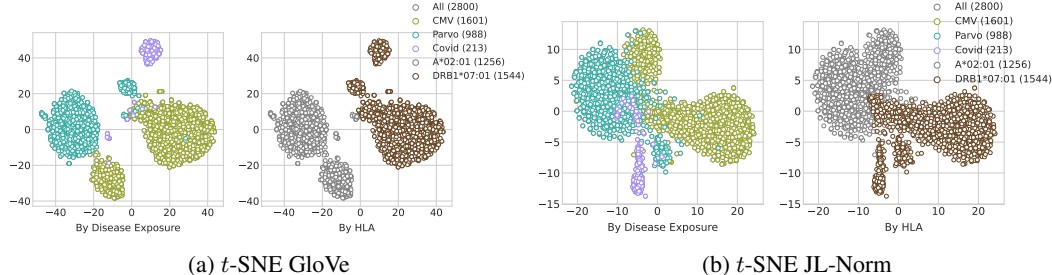

(a) $t$-SNE GloVe             (b) $t$-SNE JL-Norm

Figure 4: $t$-SNE comparisons of (a) *postprocessed* GloVe in Equation (7) and (b) JL-Norm in Equation (6) TCR embeddings, with an embedding dimension of $d = 100$, derived from $K = 65,751$ TCR measurements across $N = 31,938$ repertoires. We present the $t$-SNE plot of a subset of 2,800 TCRs, colored by their disease exposure and HLA association.

## 2.4 ADDRESSING ANISOTROPY IN TCR EMBEDDINGS

The GloVe objective in Equation (1) optimizes for *alignment*, *i.e.*, TCR pairs with high co-occurrence are encouraged to have high cosine similarity. However, previous work has shown that GloVe embeddings tend to be *anisotropic*, occupying a narrow cone in the vector space (Arora et al., 2017; Mu & Viswanath, 2018). This anisotropy is driven by varying marginal occurrences that follow a power law distribution in our case, see Figure 8. A similar phenomena has been observed for contextual embeddings (Gao et al., 2019; Ethayarajh, 2019; Li et al., 2020). Inspired by the simple yet effective postprocessing approach by Mu & Viswanath (2018), we $(i)$ center the TCR embeddings around the origin (zero mean) for each dimension $d$; and $(ii)$ remove the effect of the dominant principal components, which are correlated with marginal frequencies. Hence, we specify the final TCR embeddings $W' \in \mathbb{R}^{K \times d}$ following Mu & Viswanath (2018):

$$\hat{w}_k = w_k - \mu$$
$$w'_k = \hat{w}_k - \sum_c \langle u_c, \hat{w}_k \rangle u_c , \qquad (7)$$

where the mean vector $\mu = \frac{1}{K} \sum_k w_k$ and $\{u_1, \ldots, u_c\}$ are the dominant $c$ principal components from $\mathrm{PCA}(\hat{W})$, s.t. $c = \lceil d/100 \rceil$.

## 3 GENERATING TCR REPERTOIRE EMBEDDINGS

In the previous section, we learned vector embeddings for TCRs. Here, we are interested in doing the same for TCR *repertoires*, which contain a *subset* of TCRs in $\mathcal{T}$. We use the *postprocessed* TCR embeddings $W'$ in Equation (7) to derive low-dimensional representations of repertoires, which are useful for various downstream tasks, such as HLA and disease predictions. Given an $n$-th individual's repertoire denoted by a binary vector $x \in \{0,1\}^K$, we assign the repertoire embeddings $R_n \in \mathbb{R}^d$ to the output of the pooling function

$$R_n := \rho(x, W') = \frac{x^\top W'}{\sum_k x_k} , \qquad (8)$$

where $\sum_k x_k$ is the total number of TCRs observed in a repertoire and Equation (8) applies a *mean pooling* set transformation on the dimensions $d$ of the repertoire-specific TCR embeddings, *i.e.*, the *non-zero* entries of $x$. Mean pooling is a simple linear transformation across *repertoire-specific* TCR embedding dimensions, which enables interpretability of the repertoire embeddings. Since co-occurring TCRs cluster in specific directions in $\mathbb{R}^d$, the mean vector shall be skewed towards those directions, and can capture all the clusters when $d$ is large enough (see Section 5 for related empirical findings and discussions). While mean pooling has achieved relative success in aggregating sentence embeddings (Shen et al., 2018), alternative advanced set pooling mechanisms, such as set transformers (Lee et al., 2019) and optimal transport-based kernel (Mialon et al., 2021), could also be considered.

## 4 EXPERIMENTS

We now assess the performance of TCR and repertoire embeddings on 5 disease prediction and 145 HLA inference binary classification tasks. We benchmark the proposed JL-GLOVE algorithm against competitive baselines. Additionally, we demonstrate the scalability of the embeddings on these tasks with respect to both the increasing number of TCRs and the growing size of repertoires. PyTorch code to train JL-GLOVE can be found at `https://github.com/microsoft/jl-glove`. We summarize the training procedure in Appendix A.6.

### 4.1 REPERTOIRE DATASETS

We train JL-GLOVE embeddings using two different training cohorts: $i$) TDETECT cohort of $N = 31,938$ repertoires (May et al., 2024) and $ii$) PUBLIC cohort of $N = 3,996$ repertoires that are publicly available. *Both training datasets are unlabeled.* To demonstrate the performance on downstream tasks we use two further datasets. The MULTIID dataset is a collection of $N = 10,725$ repertoires with binary disease and HLA labels. The EMERSON dataset matches that described in Emerson et al. (2017) and has both HLA and CMV labels. See Table 1 for an overview. We consider two TCR sequence selection approaches: HLADB, which is biased for HLA associations, and an unbiased GENPROB. See Appendix A.3 for more details on TCR sequence selection.

Table 1: Summary of the MULTIID and EMERSON repertoire datasets.

| Dataset | Category | Total | COVID-19 | HSV-1 | HSV-2 | Parvo | CMV | Typed HLA |
|---------|----------|-------|----------|-------|-------|-------|-----|-----------|
| MULTIID | Train [Disease/HLA] | 6,136 | 6,135 | 847 | 872 | 876 | - | 1,640 |
| MULTIID | Test [Disease/HLA] | 4,590 | 4,590 | 220 | 225 | 204 | - | 388 |
| EMERSON | Train [Disease] | 666 | - | - | - | - | 666 | 666 |
| EMERSON | Test [Disease] | 120 | - | - | - | - | 120 | 0 |
| EMERSON | Train [HLA] | 466 | - | - | - | - | 466 | 466 |
| EMERSON | Test [HLA] | 200 | - | - | - | - | 200 | 200 |

### 4.2 QUANTITATIVE RESULTS

We validate that inferred TCR embeddings encode immune genetics and exposure history using repertoire-level classification tasks. We employ L1/L2-regularized *disease-specific logistic regression* models, parameterized by $\boldsymbol{u^m}$: $\Phi^m(\boldsymbol{u^m}; R, Y^m) : \mathcal{R}^d \to \mathcal{Y} \in \{0, 1\}$, given *shared repertoire embeddings* $R_n \in \mathcal{R}$ in Equation (8) and binary disease/HLA labels $Y_n^m \in \mathcal{Y}$ for $m = 150$; 5-disease and 145-HLA classification tasks. We tune the penalty parameter through 5-fold cross-validation.

**Classifying HLA types** Figures 5a and 5b show the area under the receiver operating characteristic curve (AUROC) performance of the repertoire embeddings on predicting 145 HLA types in the MULTIID cohort test data, matching the HLAs modeled in Zahid et al. (2024). Figure 5a demonstrates the performance improvement in HLA classification as we scale from $\approx 65,000$ to $\approx 4$ million HLA-associated TCRs. As expected the performance improves with increasing numbers of TCRs. Figure 5b repeats this experiment but scaling up the number of repertoires used from the PUBLIC cohort to the TDETECT cohort with $\approx 500,000$ TCRs chosen in an unbiased manner to be enriched for memory TCRs (see Appendix A.3). Again the performance improves with the larger repertoire set, although it is important to note that the performance of the TCRs chosen in an unbiased manner is inferior to a set of sequences specifically chosen to predict HLAs, as expected. We observe a similar trend in the EMERSON cohort test data, see Appendix Figures 10a and 10b.

**Classifying disease labels** We select HLADB sequences with $K \in \{65,751; 360,596; 3,796,900\}$, such that all sets have $\approx 30,000$ TCRs known to be associated with all our disease labels. The remaining sequences are randomly selected from HLADB to achieve the desired $K$. Hence, by construction, we expect $K = 65,751$ to provide an upper bound for our disease classification tasks, which we wish to maintain as we scale $K$. Table 2 shows the performance of the repertoire embeddings on three disease classification tasks, COVID-19, Parvo virus and CMV. We compare to two baselines: i) DeepRC, a BERT-based TCR protein sequence model (Widrich et al., 2020), and ii) ESLG, which selects disease-associated TCRs per endpoint using a case-control setup and then fits a simple linear classifier (Greissl et al., 2021; Emerson et al., 2017). In Table 3, we

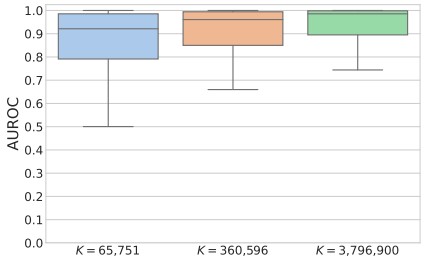
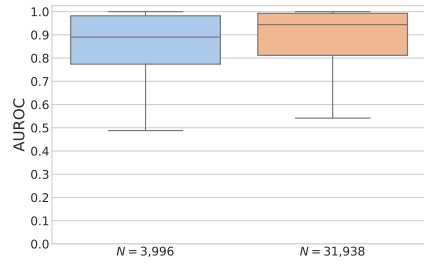

(a) varying HLADB $K$ and fixed $N = 31,938$      (b) varying $N$ and fixed GENPROB $K \approx 500,000$

Figure 5: JL-GLOVE ($d = 100$) AUROC distribution for the binary classification of 145 common HLAs, given repertoire embeddings from Equation (8) using MULTIID test data. We demonstrate the impact of scaling the number of TCRs (a) and the number of repertoires (b) in measurements of $X$.

also show the ability of the repertoire embeddings to disentangle two homologous viruses, HSV-1 and HSV-2. Here, we benchmark JL-GLOVE against ESLG and AIRIVA, a VAE-based model that uses a similar TCR sequence selection to ESLG (Pradier et al., 2023).

Although we expected a drop in JL-GLOVE's disease classification performance as we scale $K$ (at fixed $d$, see Theorem 2.1), the impact varies across disease exposures, with the most significant impact on Parvo virus and HSV. This is anticipated, given that the simple mean pooling-based repertoire aggregation method becomes more sensitive to noise as we scale $K$, thus classifiers for diseases with a stronger overall immune response, like CMV, remain more robust to increasing $K$. We aim to derive an aggregation function capable of classifying a wide range of pathogens and consider improving repertoire aggregation as key future work. We also observe that our results are sensitive to the embeddings dimension $d$ as we scale TCRs $K$ (see Figure 13). In general, we find that we need embedding dimensions $d \geq 100$ to obtain reasonable disease prediction performance. Refer to Table 4 and Table 5 for disease classification results based on unbiased GENPROB sequence selection, which is not as performant as HLADB sequences, but whose performance is invariant to the number of repertoires $N$.

Table 2: Comparison of ESLG, DeepRC, and JL-GloVe ($d = 100$; $N = 31,938$; varying HLADB TCRs $K$) disease-specific models on MULTIID and EMERSON test data sets. We report the median AUROC and sensitivity at 98% specificity, along with the 95% confidence intervals (CI) from 100 bootstrap samples.

| | Parvo | | CMV | | COVID-19 | |
|---|---|---|---|---|---|---|
| Model | Sensitivity | AUROC | Sensitivity | AUROC | Sensitivity | AUROC |
| ESLG | $0.30 \pm 0.16$ | $0.73 \pm 0.06$ | $0.63 \pm 0.38$ | $0.93 \pm 0.01$ | $0.70 \pm 0.06$ | $0.95 \pm 0.04$ |
| DeepRC (Widrich et al., 2020) | - | - | - | $0.83 \pm 0.002$ | - | - |
| JL-GloVe ($K = 65,751$) | $0.48 \pm 0.26$ | $0.85 \pm 0.05$ | $0.95 \pm 0.50$ | $0.99 \pm 0.02$ | $0.86 \pm 0.03$ | $0.97 \pm 0.01$ |
| JL-GloVe ($K = 360,596$) | $0.30 \pm 0.14$ | $0.76 \pm 0.07$ | $0.98 \pm 0.68$ | $0.98 \pm 0.03$ | $0.81 \pm 0.04$ | $0.96 \pm 0.01$ |
| JL-GloVe ($K = 3,796,900$) | $0.01 \pm 0.07$ | $0.53 \pm 0.10$ | $0.39 \pm 0.49$ | $0.92 \pm 0.05$ | $0.59 \pm 0.06$ | $0.92 \pm 0.01$ |

## 5 INTERPRETING THE GEOMETRY OF TCR AND REPERTOIRE EMBEDDINGS

Neural word embeddings learned via algorithms such as GloVe and Word2Vec are known to preserve contextual semantic properties through linear vector arithmetic, *i.e.*, the word embeddings for "$queen \approx king - man + woman$" (Mikolov et al., 2013b; Levy & Goldberg, 2014). Further, the GloVe objective function in Equation (1) encourages TCRs that co-occur across repertoires, relative to their overall occurrence counts to have larger dot products. Essentially, they will have higher cosine similarity (*i.e.,* point in similar directions to each other in the embedding space) and may have larger embedding norms.

**TCR embeddings cluster by antigen**    Figure 4a illustrates that the TCR embeddings learned by minimizing Equation (1) indeed capture the disease exposure and HLA of a TCR. Thus we expect that TCRs responding to the same antigen should have high cosine similarity, *i.e.*, point in similar

Table 3: Comparison of HSV disease models on MULTIID test data. We report the AUROC and sensitivity at 98% specificity, both overall and stratified by subtype. We present the median and 95% CI from 100 bootstrap samples for the models AIRIVA and ESLG from Pradier et al. (2023), and JL-GloVe ($d = 100; N = 31,938$; varying HLADB TCRs $K$).

| HSV-1 Model | Overall | | HSV-2 negative | | HSV-2 positive | |
|---|---|---|---|---|---|---|
| | Sensitivity | AUROC | Sensitivity | AUROC | Sensitivity | AUROC |
| ESLG (Pradier et al., 2023) | $0.12 \pm 0.10$ | $0.62 \pm 0.09$ | $0.18 \pm 0.15$ | $0.63 \pm 0.12$ | $0.14 \pm 0.17$ | $0.50 \pm 0.19$ |
| AIRIVA (Pradier et al., 2023) | $0.30 \pm 0.12$ | $0.74 \pm 0.09$ | $0.35 \pm 0.20$ | $0.74 \pm 0.10$ | $0.32 \pm 0.22$ | $0.67 \pm 0.16$ |
| JL-GloVe ($K = 65,751$) | $051 \pm 0.11$ | $0.90 \pm 0.04$ | $0.58 \pm 0.16$ | $0.92 \pm 0.05$ | $0.37 \pm 0.17$ | $0.78 \pm 0.12$ |
| JL-GloVe ($K = 360,596$) | $0.46 \pm 0.15$ | $0.87 \pm 0.05$ | $0.57 \pm 0.14$ | $0.91 \pm 0.05$ | $0.40 \pm 0.22$ | $0.81 \pm 0.11$ |
| JL-GloVe ($K = 3,796,900$) | $0.09 \pm 0.11$ | $0.64 \pm 0.09$ | $0.03 \pm 0.09$ | $0.64 \pm 0.12$ | $0.27 \pm 0.19$ | $0.65 \pm 0.14$ |

(a) HSV-1 Prediction Task

| HSV-2 Model | Overall | | HSV-1 negative | | HSV-1 positive | |
|---|---|---|---|---|---|---|
| | Sensitivity | AUROC | Sensitivity | AUROC | Sensitivity | AUROC |
| ESLG (Pradier et al., 2023) | $0.11 \pm 0.10$ | $0.75 \pm 0.07$ | $0.16 \pm 0.21$ | $0.79 \pm 0.16$ | $0.12 \pm 0.15$ | $0.75 \pm 0.10$ |
| AIRIVA (Pradier et al., 2023) | $0.37 \pm 0.18$ | $0.78 \pm 0.10$ | $0.57 \pm 0.26$ | $0.86 \pm 0.12$ | $0.32 \pm 0.20$ | $0.77 \pm 0.10$ |
| JL-GloVe ($K = 65,751$) | $0.37 \pm 0.22$ | $0.90 \pm 0.04$ | $0.21 \pm 0.73$ | $0.98 \pm 0.04$ | $0.30 \pm 0.18$ | $0.87 \pm 0.06$ |
| JL-GloVe($K = 360,596$) | $0.25 \pm 0.16$ | $0.86 \pm 0.05$ | $0.21 \pm 0.71$ | $0.97 \pm 0.05$ | $0.23 \pm 0.13$ | $0.82 \pm 0.07$ |
| JL-GloVe ($K = 3,796,900$) | $0.08 \pm 0.10$ | $0.72 \pm 0.07$ | $0.07 \pm 0.22$ | $0.77 \pm 0.12$ | $0.12 \pm 0.14$ | $0.70 \pm 0.09$ |

(b) HSV-2 Prediction Task

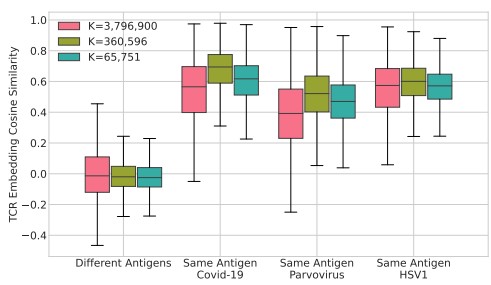

(a) TCR embedding cosine similarity by antigen

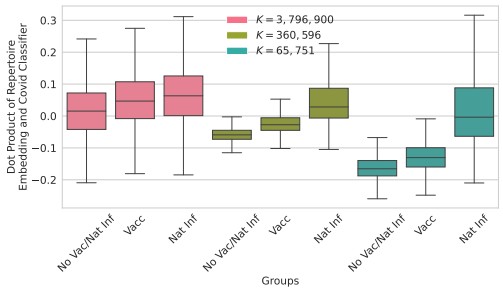

(b) COVID-19 classifier logits by subgroup

Figure 6: JL-GLOVE ($d = 100; N = 31,938$; varying HLADB TCRs $K$) distribution of (a) cosine similarity calculated between TCR embeddings associated to the same antigen *vs.* different antigens and (b) the dot products $\langle \boldsymbol{u}^m, R_n \rangle$ between the $\boldsymbol{u}^m$ weights of the MULTIID COVID-19 classifier and TDETECT cohort repertoire embedded vectors $R_n$.

directions in the TCR embedding space. We directly test the claim with TCR antigen associations derived using a multiplexed identification of T cell receptor antigen specificity (MIRA) (Klinger et al., 2015) assay, specifically designed to identify antigen specific TCRs. The MIRA assay (see A.7 for details) has been used to associate millions of TCRs to thousands of antigens from known pathogens. Here, we limit the analysis to SARS-CoV-2, Parvo virus, and HSV-1 antigens with $> 4$ associated TCRs. As expected, Figure 6a shows that TCRs associated with the same antigen have significantly higher cosine similarity, *i.e.*, point in similar directions, than TCRs associated with different antigens. Importantly, this property is preserved as we scale the number of TCRs $K$ and the number of repertoires $N$ (see Figure 11a).

**Classifier weights stratify repertoires by antigen**   Using previously developed models, we can predict with high precision and accuracy whether a subject in our sample has had a natural SARS-CoV-2 infection, vaccination but no natural infection or neither (Pradier et al., 2023). Subjects with natural infection respond to a broad range of proteins derived from the SARS-CoV-2. In contrast, vaccines use only the spike protein and therefore vaccinated subjects elicit T cell response to a subset of all SARS-CoV-2 antigens derived from the spike protein. Figure 6b shows the distribution of the dot product of the weights of the MULTIID COVID-19 classifier and TDETECT cohort repertoire embeddings $\langle \boldsymbol{u}^m, R_n \rangle$. Interestingly, the dot product is largest for subjects with natural infection, followed by subjects who have been vaccinated and smallest for subjects who are neither vaccinated or previously infected. This result indicates that the weights of the MULTIID COVID-19 classifier

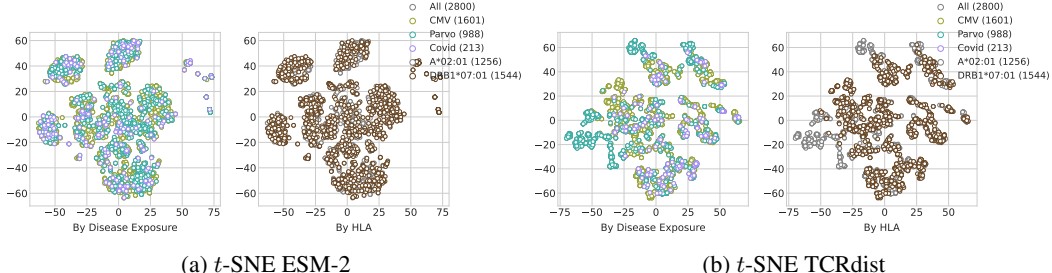

(a) $t$-SNE ESM-2              (b) $t$-SNE TCRdist

Figure 7: $t$-SNE comparisons of (a) ESM-2 (Rives et al., 2021) and (b) TCRdist (hamming) (Dash et al., 2017) embeddings derived from a subset of 2,800 disease- and HLA-associated TCR amino acid sequences. We present the $t$-SNE plot of a subset of TCRs, colored by their disease exposure and HLA association.

generalize to the TDETECT cohort and represent a superposition of vectors that point in the direction of various antigens derived from the SARS-CoV-2 genome. The broader the immune response in terms of antigens, the greater the dot product with the classifier. However, we notice a smaller separation at $K = 3,796,900$, which we attribute to the lack of robustness of the mean pooling function for large $K$ and the need for $d > 100$. See Figure 11b for additional results on unbiased GENPROB sequences, demonstrating poorer repertoire stratification than HLADB sequences.

**Comparisons to TCR protein sequence embedding approaches** Embedding TCRs and repertoires using TCR co-occurrence yields interpretable and biologically relevant geometric properties which are not captured by other embedding schemes. Figure 7 demonstrates that TCR embeddings derived from protein models such as the pretrained ESM-2 (Rives et al., 2021) and protein distance based approach TCRdist (Dash et al., 2017) fail to cluster the embedding space by HLA or disease exposure unlike our proposed JL-GLOVE co-occurrence based TCR embeddings shown in Figure 4. These results are not surprising since TCR amino acid sequences are randomly generated via the random V(D)J recombination (Tonegawa, 1983), which violates evolutionary assumptions made in protein models such as ESM-2. Consistent with findings from Nagano et al. (2024), the quality of the TCRdist embeddings is slightly better than ESM-2. However, TCRdist assumes that TCR proteins binding to the same antigen often share amino acid sequence similarity, *i.e*, small hamming or BLOSUM distance, which does not generalize across all pathogens. Also, the dimensions of TCRdist embeddings increase with the number of TCRs, since $d = K$, and computational complexity scales quadratically with the number of TCRs because the embeddings are derived from $K^2$ comparisons.

## 6   CONCLUSIONS

Most data are unstructured, sparse, and heterogeneous, and representing such data is a primary challenge for modeling. Here, we generate low-dimensional representations of TCRs and TCR repertoires based on the co-occurrence of TCRs at scale. We propose JL-GLOVE, which employs the GloVe algorithm to learn immunologically meaningful TCR embeddings. Moreover, we improve GloVe's computational efficiency in terms of memory usage and training time by leveraging a powerful initialization based on random projection theory. Further, the proposed JL-GLOVE algorithm is general and can be repurposed to learn embeddings in other data modalities where alignment is derived from co-occurrence statistics.

Extensive experimental results show that the repertoire embeddings summarize the immune genetics and pathogenic exposure histories of individuals. Notably, TCR embeddings cluster by antigens, and this property remains invariant as we scale the number of TCRs and repertoires. Moreover, as the amount of TCR and repertoire data increases, these embeddings will continue to improve, enabling the quantification of more disease exposures for rarer HLA types. These embeddings can be combined with other modalities, such as single-cell RNA sequencing, to provide more information for individual T cells, and with clinical modalities, such as electronic health records, to offer more subject-level information. Ultimately, personalized medicine and individualized treatments will require a careful accounting of immune genetics and pathogenic exposure history.

ACKNOWLEDGMENTS

The authors would like to thank the anonymous reviewers for their insightful comments. This research was conducted as part of a joint collaboration between Adaptive Biotechnologies and Microsoft Research.

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

# A APPENDIX

## A.1 JL NORMALIZATION PROOF

*Proof.* We adapt the following proof from the discussions in Section 4 of Roughgarden & Valiant (2024). Let us define

$$\hat{\boldsymbol{w}}_k^{\mathrm{JL}}(m) \coloneqq \sum_{j=1}^{K} \boldsymbol{c}_k(j) P(j, m) \,,$$

where $m \in \{1, 2, \ldots, d\}$ and the JL transform $\boldsymbol{w}_k^{\mathrm{JL}}(m) = \frac{1}{\sqrt{d}} \hat{\boldsymbol{w}}_k^{\mathrm{JL}}(m)$, see Equation (2). Note $\hat{\boldsymbol{w}}_k^{\mathrm{JL}}(m)$ is a zero-mean random variable with variance

$$\sum_{j} \boldsymbol{c}_k(j)^2 = \|\boldsymbol{c}_k\|^2 \,,$$

hence $\mathbb{E}[\hat{\boldsymbol{w}}_k^{\mathrm{JL}}(m)^2] = \|\boldsymbol{c}_k\|^2$. Consequently, we have

$$\begin{aligned}
\left\|\boldsymbol{w}_k^{\mathrm{JL}}\right\|^2 &= \sum_{m=1}^{d} \boldsymbol{w}_k^{\mathrm{JL}}(m)^2 \\
&= \frac{1}{d} \sum_{m} \hat{\boldsymbol{w}}_k^{\mathrm{JL}}(m)^2 \,.
\end{aligned} \tag{9}$$

Note that Equation (9) is an average of $d$ unbiased estimators of $\|\boldsymbol{c}_k\|^2$. By central-limit theorem, we have

$$\left\|\boldsymbol{w}_k^{\mathrm{JL}}\right\|^2 \xrightarrow{p} \|\boldsymbol{c}_k\|^2 \tag{10}$$

Finally, we note

$$\tilde{\boldsymbol{w}}_k^{\mathrm{JL}} = \frac{1}{\sqrt{d}} \langle \tilde{\boldsymbol{c}}_k, P \rangle = \frac{1}{\sqrt{d}} \frac{\langle \boldsymbol{c}_k, P \rangle}{\|\boldsymbol{c}_k\|} = \frac{\boldsymbol{w}_k^{\mathrm{JL}}}{\|\boldsymbol{c}_k\|} \approx \frac{\boldsymbol{w}_k^{\mathrm{JL}}}{\left\|\boldsymbol{w}_k^{\mathrm{JL}}\right\|} \,. \tag{11}$$

The last step in Equation (11) is due to Equation (10) for large $d$. Specifically, one can first compute $W^{\mathrm{JL}}$ following Equation (2) in linear-time, and then normalize the rows of $W^{\mathrm{JL}}$ to get $\tilde{W}^{\mathrm{JL}}$. This approach is approximately equivalent to the quadratic-time alternative of computing and normalizing $C$ before projecting via $P$ to obtain $\tilde{W}^{\mathrm{JL}}$ using Equation (6). □

## A.2 ACCOUNTING FOR VARYING MARGINAL OCCURRENCES OF TCRS

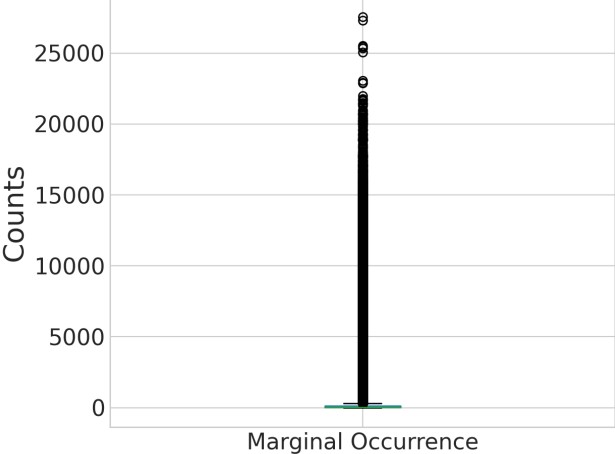

Figure 8: Marginal occurrence counts of $K = 3,796,900$ TCRs computed from $N = 31,938$ repertoires.

It is reasonable to expect $\|c_i - c_j\|$ to be smaller when TCRs $t_i$ and $t_j$ frequently co-occur, and larger otherwise. However, this can be misleading due to the different marginal occurrence frequencies of TCRs, which are *power-law distributed in our case*; see Figure 8. Consider the following illustrative co-occurrence matrix $C$ of 4 TCRs:

$$C = \begin{bmatrix} 6 & 0 & 0 & 0 \\ 0 & 8 & 0 & 0 \\ 0 & 0 & 75 & 60 \\ 0 & 0 & 60 & 80 \end{bmatrix} \underbrace{\Longrightarrow}_{\substack{\text{Unit-normalize} \\ \text{the rows}}} \tilde{C} = \begin{bmatrix} 1 & 0 & 0 & 0 \\ 0 & 1 & 0 & 0 \\ 0 & 0 & 0.78 & 0.63 \\ 0 & 0 & 0.6 & 0.8 \end{bmatrix}. \tag{12}$$

Note that $t_1$ and $t_2$ never co-occur, while $t_3$ and $t_4$ co-occur frequently. Despite this, we have $\|c_1 - c_2\| = 10$ which is smaller than $\|c_3 - c_4\| = 25$. The main reason is that $c_1$ and $c_2$ have smaller norms than $c_3$ and $c_4$ because $t_1$ and $t_2$ are rarer TCRs. This observation suggests that we should remove the effect of the TCR embeddings' *norms* from the pairwise L2 distances across $c_i$ before leveraging the random projection theory for dimensionality reduction (see Theorem 2.1).

### A.3 TCR Sequence selection

The probability of a specific TCR (amino acid sequence) being generated varies by $\sim$20 orders of magnitude ranging for $10^{-6} - 10^{-30}$ (Sethna et al., 2019). Thus, TCRs with high likelihood of random generation will necessarily co-occur with one another. We are only interested in modeling co-occurrence of TCRs that are antigen-experienced (*i.e.*, memory TCRs) as these TCRs carry meaningful information about immune genetics and pathogenic exposure history. We employ multiple selection strategies to enrich our sample for memory TCRs. Zahid et al. (2024) use labels of HLAs to identify sequences that have strong statistical association to HLAs, meaning they are likely memory TCRs. We refer to these sequences as HLADB. Refer to May et al. (2024) for more details of how the set used here is derived.

We also select a set of TCRs enriched for memory in an unbiased manner requiring no labels using a combination of the TCR generation probability and the observed frequency in repertoires. The generation probability provides a naive prior on the expected frequency; TCRs that are memory undergo clonal expansion and therefore will likely be observed at a higher frequency in repertoires than the naive expectation from the generation probability (DeWitt III et al., 2018). We empirically determine the observed frequency cutoff as a function of generation probability using our set of sequences from HLADB. In other words, using HLADB which is a set of sequences known to be enriched for memory TCRs, we derive the distribution of observed frequencies for those sequences relative to their naive expectation frequency given by the generation probability and derive a cutoff. We then use this cutoff to select from all TCRs in our repertoires which does not require any repertoire level labels. We refer to these sequences as GENPROB.

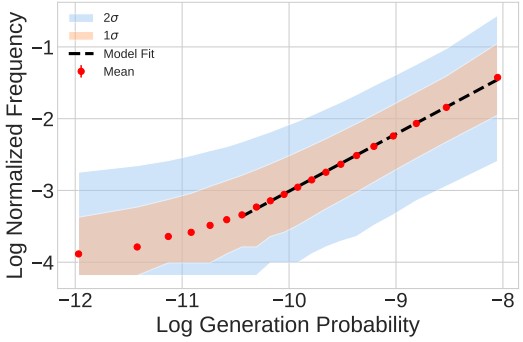

Figure 9: Normalized frequency of 4 million HLADB TCRs as a function of generation probability. Red points show the mean normalized frequency in equally populated bins of generation probability. Orange and blue shaded regions indicate the central limits containing 68% ($1\sigma$) and 95% ($2\sigma$) of the data, respectively. The black line is fit to the data in the range indicated according to Equation (13). The deviation of the normalized frequency from a power-law at low generation probability is a consequence of sample size; given a larger number of repertoires, the power law relationship would extend to lower values of generation probability.

Figure 9 shows the distribution of normalized frequency of TCRs as a function of generation probability. The mean normalized frequency as a function of generation probability is a power law which is fit by a line (black curve in Figure 9) in log-log space given by

$$\log_{10} f_{\text{obs}}(P_{\text{gen}}) = m(\log_{10} P_{\text{gen}} + 9) + c, \tag{13}$$

where $m = 0.797$ is the slope and $c = -2.213$ is the intercept which is defined as the mean at a generation probability of $10^{-9}$ (*i.e.*, $\log_{10} P_{\text{gen}} + 9$); $\log_{10} f_{\text{obs}}$ and $\log_{10} P_{\text{gen}}$ are the logarithm base 10 of the normalized frequency of TCRs (*i.e.*, fraction of repertoires in which the TCR is observed) and logarithm base 10 of the TCR amino acid sequence generation probability calculated using OLGA (Sethna et al., 2019), respectively. Let $Y(P_{\text{gen}}) := \log_{10} f_{\text{obs}}(\cdot)$ denote the value of $\log_{10} f_{\text{obs}}(\cdot)$ at a *fixed generation probability* $P_{\text{gen}}$. We fit $Y(P_{\text{gen}})$ with a Gaussian distribution with mean $\mu = \log_{10} f_{\text{obs}}(P_{\text{gen}})$ and standard deviation $\sigma = 0.473$, which is constant and independent of generation probability (see shaded regions in Figure 9). The full distribution is defined by our power-law fit, which defines the Gaussian mean, and the invariant standard deviation. Thus, we use the quantile function to select sequences at a fixed percentile probability $\tau$, *s.t.* $\tilde{c} = \inf\{\tilde{c} \in \mathbb{R} : P\left(Y(P_{\text{gen}} = 10^{-9}) < \tilde{c}\right) = \tau\}$. Specifically, to select sequences at a specific percentile probability $\tau$, we set $c \leftarrow \tilde{c}$ in Equation (13). This yields an observed frequency threshold that increases with generation probability and is at a fixed percentile $\tilde{c}$ of the empirical distribution of HLADB TCRs, as observed in TDETECT samples. Note, when $\tau = 0.5$, then $\tilde{c} = c$, and we recover the mean fit shown by the black line in Figure 9. We choose $\tau$ to yield approximately $500,000$ sequences in the TDETECT and PUBLIC cohorts. This selection requires us to set $1 - \tau$ to 0.015 and 0.085 for TDETECT and PUBLIC cohorts, respectively.

### A.4 COMPUTING TCR CO-OCCURRENCES

Generating $C$ is computationally expensive, and although $C$ is sparse, as the number of TCRs $K$ grows, it can become very large. Thus, computing $X^\top X$ in one step can become intractable. Fortunately, this operation is easily parallelized with Spark (Zaharia et al., 2010) by distributing the computation along either rows or columns of $X$, as shown in Algorithm 1.

---

**Algorithm 1** Computing TCR co-occurrences $C$

---

**Input**: TCR repertoire binary matrix $X$
**Output**: TCR co-occurrences $C$

1: **while** $\beta \leq K$ **do**
2:     Given the $n$-th individual's repertoire $\boldsymbol{x}_n$, filter $\boldsymbol{x}_n$ to generate $\hat{\boldsymbol{x}}_n$, which represents the presence or absence of the subset of TCRs $\hat{\mathcal{T}} = \{t_\alpha, \ldots, t_\beta\}\ \forall n\ \ s.t.\ \ \beta - \alpha = 100$ and $\hat{\mathcal{T}} \subseteq \mathcal{T}$
3:     Join $\hat{\boldsymbol{x}}_n$ and $\boldsymbol{x}_n$ on $n$ to generate all TCR pairs $t_i$ in $\mathcal{T}$ and $t_j$ in $\hat{\mathcal{T}}$
4:     Group by $(i, j)$ across all $n$ to create $\hat{C}_\alpha$
5: **end while**
6: Union all $\hat{C}_\alpha$ to create $C$
7: **return** $C$

---

### A.5 COMPUTING JL EMBEDDINGS

Algorithm 2 provides the sparse matrix algorithm for computing the JL-Norm embeddings $\tilde{W}^{\text{JL}}$, which are used to initialize GloVe embeddings in Equation (1).

### A.6 TRAINING JL-GLOVE

We initialize the TCR embeddings $W$ with the JL-Norm embeddings $\tilde{W}^{\text{JL}}$ and use the Adagrad optimizer (Duchi et al., 2011) with learning rate 0.05 to minimize the GloVe objective in Equation (1) via stochastic gradient descent on minibatches from $C$. We leverage the distributed Dask framework [1] to load minibatches (partitions) of $C$, *i.e.*, millions of entries in $C$ stored as Parquet files [2]. Further,

---

[1] https://www.dask.org/
[2] https://parquet.apache.org/

---

**Algorithm 2** (Approximately) Computing JL-Norm embeddings $\tilde{W}^{\text{JL}}$

---

**Input**: TCR repertoire binary matrix $X$

**Output**: Approximation to $\tilde{W}^{\text{JL}}$ (see Proposition 2.2)

  1: Construct the projection matrix $P$ by sampling its elements i.i.d. following equation 3.
  2: Compute $M_1 = XP$
  3: Compute $M_2 = X^{\mathsf{T}} M_1$
  4: Compute $M = \frac{1}{\sqrt{d}} M_2$ (see equation 2)
  5: $\boldsymbol{m}_i \leftarrow \frac{\boldsymbol{m}_i}{\|\boldsymbol{m}_i\|}$, for all rows $\boldsymbol{m}_i$ of $M$
  6: **return** $M$

---

we use PyTorch Lightning [3] for distributed data parallel training across one node equipped with 4 NVIDIA A100 80GB GPUs (if $K \geq 500,000$) or 2 Tesla V100 16GB GPUs (if $K < 500,000$).

### A.7 MIRA ASSAY DESCRIPTION

In brief, a panel of antigens associated with specific proteins are presented on HLAs in specific wells. T cells taken from a blood draw are separated and put into the solution. T cells that respond to these antigens are activated. Activated T cells are identified via surface proteins and sorted out for TCR sequencing. Different subsets of antigens are present in different wells and using combinatoric reconstruction, specific T cells are associated with specific antigens.

### A.8 SCALING LAWS

Figure 10a and Figure 10b demonstrate that quality of the embeddings improves as we scale the number of TCRs $K$ and repertoires $N$. We quantify the scaling impact with AUROC distribution across 145 HLAs on the EMERSON cohort test data.

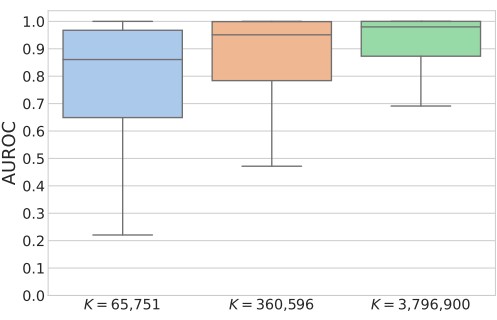
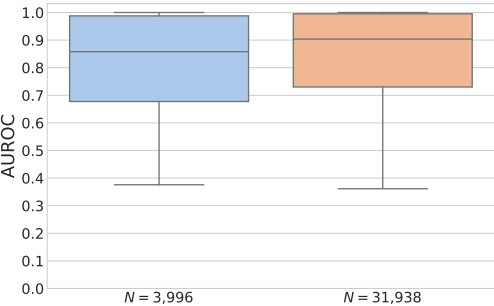

    (a) varying HLADB $K$ and fixed $N = 31,938$      (b) varying $N$ and fixed GENPROB $K \approx 500,000$

Figure 10: JL-GLOVE ($d = 100$) AUROC distribution of the binary classification of 145 common HLAs, given repertoire embeddings from Equation (8) using EMERSON test data. We demonstrate the impact of scaling the number of TCRs (a) and the number of repertoires (b) measurements in $X$.

### A.9 INTERPRETING THE GEOMETRY OF TCR AND REPERTOIRE EMBEDDINGS

Figure 11a shows that GENPROB TCRs associated with the same antigen have significantly higher cosine similarity, *i.e.*, they point in a similar direction than TCRs associated with different antigens. However, we observe lower cosine similarities for similar antigens when the number of repertoires $N$ is lower.

Next, we inspect the dot products between the COVID-19 logistic regression classifiers' weights and TCR embeddings $\langle \boldsymbol{u}^m, R_n \rangle$. Figure 11b does not show the same separation observed in Figure 6b in terms of the dot product, between subjects with natural infection, vaccinated individuals, and

---

[3]https://lightning.ai/

those who are neither vaccinated nor previously infected. We attribute this discrepancy to: $i$) the less performant GENPROB TCR sequence selection, resulting in a weak COVID-19 classifier $\boldsymbol{u}^m$ (see Table 4); and $ii$) the use of a simple mean pooling repertoire aggregation function, which is not robust to noise when $K$ is too large. Additionally, we find that the TCRs known to be associated with disease/HLA labels (*i.e.,* enhanced sequences) have significantly larger dot products with the corresponding classifier's weights. We depict this in Figure 12.

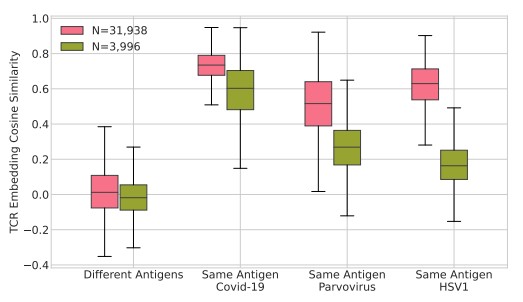 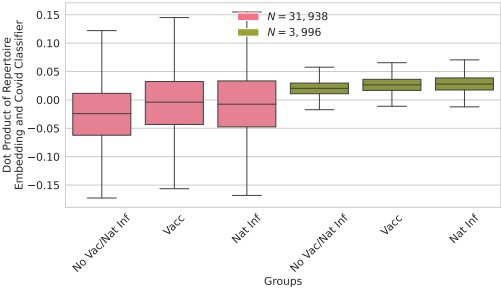

(a) TCR embedding cosine similarity by antigen     (b) COVID-19 classifier logits by subgroup

Figure 11: JL-GLOVE ($d = 100$; $K \approx 500,000$ GENPROB TCRs; varying $N$ repertoires) distribution of (a) cosine similarity calculated between TCR embeddings associated to the same antigen *vs.* different antigens and (b) the dot products $\langle \boldsymbol{u}^m, R_n \rangle$ between the $\boldsymbol{u}^m$ weights of the MULTIID COVID-19 classifier and TDETECT cohort repertoire embedded vectors $R_n$.

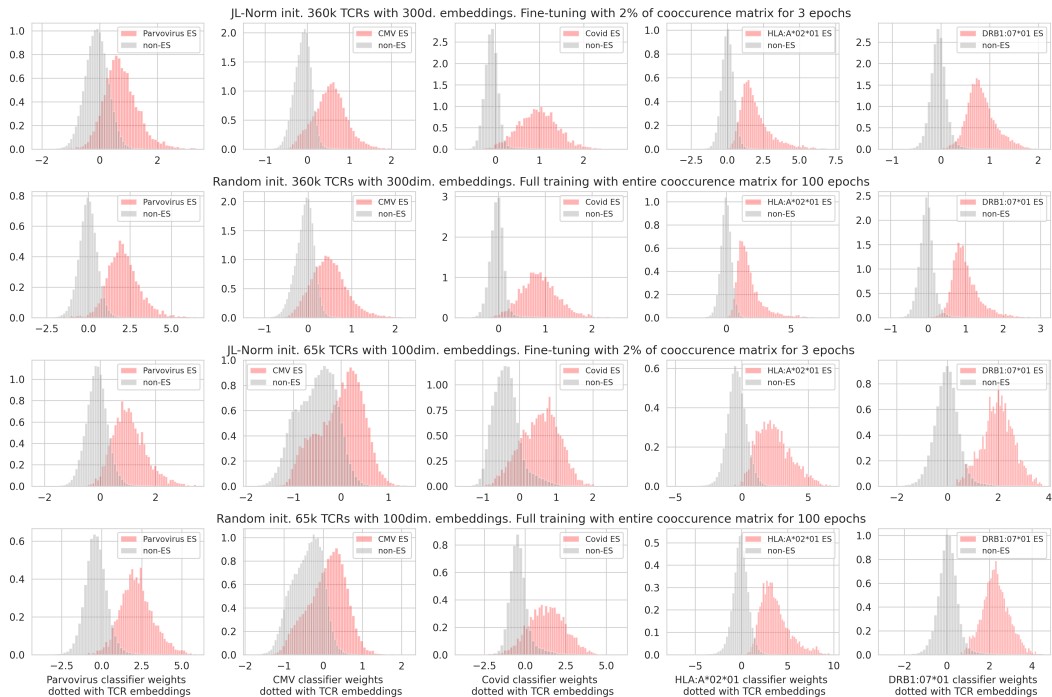

Figure 12: Logistic regression classifier weights dotted with TCR embeddings $\langle \boldsymbol{u}^m, \boldsymbol{w}_k \rangle$, respectively. TCRs that are *enhanced sequences* (ES) for the corresponding disease/HLA labels point in the direction given by the classifier's weight, *i.e.*, they yield higher cosine similarity.

## A.10 QUANTITATIVE CLASSIFICATION RESULTS

We present additional quantitative classification results for JL-GloVe ($d = 100$; GENPROB $K \approx 500,000$; varying $N$) in Table 4 and Table 5. Compared to Table 6 and Table 7, we note that the

importance of TCR selection $K$ is greater than that of repertoire measurements $N$ used to train JL-GloVe on downstream classification tasks. Further, Figure 13 highlights that weaker signals, such as the Parvo virus, are highly sensitive to the embedding dimensions $d$ as we scale TCRs $K$; *i.e.*, we need higher embedding dimensions to capture such signals using a simple mean pooling-based repertoire embedding aggregation function.

Table 4: Comparison of ESLG, DeepRC, and JL-GloVe ($d = 100$; GENPROB $K \approx 500,000$; varying $N$) disease-specific models on MULTIID and EMERSON test data sets. We report the median AUROC, AUPRC, and sensitivity at 98% specificity, along with the 95% confidence intervals (CI) from 100 bootstrap samples.

| | Parvo | | | CMV | | | COVID-19 | | |
|---|---|---|---|---|---|---|---|---|---|
| Model | Sensitivity | AUROC | AUPRC | Sensitivity | AUROC | AUPRC | Sensitivity | AUROC | AUPRC |
| ESLG | $0.30 \pm 0.16$ | $0.73 \pm 0.06$ | $0.86 \pm 0.05$ | $0.63 \pm 0.38$ | $0.93 \pm 0.01$ | $0.88 \pm 0.08$ | $0.70 \pm 0.06$ | $0.95 \pm 0.04$ | $0.84 \pm 0.03$ |
| DeepRC (Widrich et al., 2020) | - | - | - | - | $0.83 \pm 0.002$ | - | - | - | - |
| JL-GloVe ($N = 3,996$) | $0.08 \pm 0.08$ | $0.58 \pm 0.09$ | $0.77 \pm 0.04$ | $0.11 \pm 0.19$ | $0.71 \pm 0.09$ | $0.66 \pm 0.07$ | $0.47 \pm 0.04$ | $0.86 \pm 0.02$ | $0.64 \pm 0.02$ |
| JL-GloVe ($N = 31,938$) | $0.02 \pm 0.09$ | $0.49 \pm 0.09$ | $0.73 \pm 0.04$ | $0.64 \pm 0.39$ | $0.93 \pm 0.05$ | $0.92 \pm 0.03$ | $0.35 \pm 0.05$ | $0.82 \pm 0.02$ | $0.53 \pm 0.02$ |

Table 5: Comparison of HSV disease models on MULTIID test data. We report the AUROC, AUPRC, and sensitivity at 98% specificity, both overall and stratified by subtype. We present the median and 95% CI from 100 bootstrap samples for the models AIRIVA and ESLG from Pradier et al. (2023), and JL-GloVe ($d = 100$; GENPROB $K \approx 500,000$; varying $N$).

| | Overall | | | HSV-2 negative | | | HSV-2 positive | | |
|---|---|---|---|---|---|---|---|---|---|
| HSV-1 Model | Sensitivity | AUROC | AUPRC | Sensitivity | AUROC | AUPRC | Sensitivity | AUROC | AUPRC |
| ESLG (Pradier et al., 2023) | $0.12 \pm 0.10$ | $0.62 \pm 0.09$ | - | $0.18 \pm 0.15$ | $0.63 \pm 0.12$ | - | $0.14 \pm 0.17$ | $0.50 \pm 0.19$ | - |
| AIRIVA (Pradier et al., 2023) | $0.30 \pm 0.12$ | $0.74 \pm 0.09$ | - | $0.35 \pm 0.20$ | $0.74 \pm 0.10$ | - | $0.32 \pm 0.22$ | $0.67 \pm 0.16$ | - |
| JL-GloVe ($N = 3,996$) | $0.06 \pm 0.25$ | $0.68 \pm 0.08$ | $0.82 \pm 0.04$ | $0.21 \pm 0.09$ | $0.70 \pm 0.09$ | $0.83 \pm 0.04$ | $0.00 \pm 0.38$ | $0.69 \pm 0.19$ | $0.83 \pm 0.83$ |
| JL-GloVe ($N = 31,938$) | $0.15 \pm 0.15$ | $0.75 \pm 0.07$ | $0.85 \pm 0.03$ | $0.09 \pm 0.16$ | $0.79 \pm 0.09$ | $0.85 \pm 0.04$ | $0.00 \pm 0.38$ | $0.67 \pm 0.18$ | $0.85 \pm 0.07$ |

(a) HSV-1 Prediction Task

| | Overall | | | HSV-1 negative | | | HSV-1 positive | | |
|---|---|---|---|---|---|---|---|---|---|
| HSV-2 Model | Sensitivity | AUROC | AUPRC | Sensitivity | AUROC | AUPRC | Sensitivity | AUROC | AUPRC |
| ESLG (Pradier et al., 2023) | $0.11 \pm 0.10$ | $0.75 \pm 0.07$ | - | $0.16 \pm 0.21$ | $0.79 \pm 0.16$ | - | $0.12 \pm 0.15$ | $0.75 \pm 0.10$ | - |
| AIRIVA (Pradier et al., 2023) | $0.37 \pm 0.18$ | $0.78 \pm 0.10$ | - | $0.57 \pm 0.26$ | $0.86 \pm 0.12$ | - | $0.32 \pm 0.20$ | $0.77 \pm 0.10$ | - |
| JL-GloVe ($N = 3,996$) | $0.07 \pm 0.14$ | $0.74 \pm 0.07$ | $0.57 \pm 0.06$ | $0.14 \pm 0.24$ | $0.77 \pm 0.15$ | $0.62 \pm 0.12$ | $0.02 \pm 0.11$ | $0.70 \pm 0.08$ | $0.54 \pm 0.06$ |
| JL-GloVe ($N = 31,938$) | $0.17 \pm 0.11$ | $0.74 \pm 0.06$ | $0.62 \pm 0.05$ | $0.09 \pm 0.17$ | $0.68 \pm 0.16$ | $0.47 \pm 0.11$ | $0.19 \pm 0.12$ | $0.78 \pm 0.08$ | $0.67 \pm 0.06$ |

(b) HSV-2 Prediction Task

Table 6: Comparison of ESLG, DeepRC, and JL-GloVe ($d = 100$; $N = 31,938$; varying HLADB TCRs $K$) disease-specific models on MULTIID and EMERSON test data sets. We report the median AUROC, AUPRC, and sensitivity at 98% specificity, along with the 95% confidence intervals (CI) from 100 bootstrap samples.

| | Parvo | | | CMV | | | COVID-19 | | |
|---|---|---|---|---|---|---|---|---|---|
| Model | Sensitivity | AUROC | AUPRC | Sensitivity | AUROC | AUPRC | Sensitivity | AUROC | AUPRC |
| ESLG | $0.30 \pm 0.16$ | $0.73 \pm 0.06$ | $0.86 \pm 0.05$ | $0.63 \pm 0.38$ | $0.93 \pm 0.01$ | $0.88 \pm 0.08$ | $0.70 \pm 0.06$ | $0.95 \pm 0.04$ | $0.84 \pm 0.03$ |
| DeepRC (Widrich et al., 2020) | - | - | - | - | $0.83 \pm 0.002$ | - | - | - | - |
| JL-GloVe ($K = 65,751$) | $0.48 \pm 0.26$ | $0.85 \pm 0.05$ | $0.94 \pm 0.01$ | $0.95 \pm 0.50$ | $0.99 \pm 0.02$ | $0.98 \pm 0.02$ | $0.86 \pm 0.03$ | $0.97 \pm 0.01$ | $0.92 \pm 0.01$ |
| JL-GloVe ($K = 360,596$) | $0.30 \pm 0.14$ | $0.76 \pm 0.07$ | $0.90 \pm 0.02$ | $0.98 \pm 0.68$ | $0.98 \pm 0.03$ | $0.96 \pm 0.03$ | $0.81 \pm 0.04$ | $0.96 \pm 0.01$ | $0.90 \pm 0.01$ |
| JL-GloVe ($K = 3,796,900$) | $0.01 \pm 0.07$ | $0.53 \pm 0.10$ | $0.73 \pm 0.04$ | $0.39 \pm 0.49$ | $0.92 \pm 0.05$ | $0.88 \pm 0.06$ | $0.59 \pm 0.06$ | $0.92 \pm 0.01$ | $0.75 \pm 0.02$ |

## A.11 COMPARISONS OF GLOVE INITIALIZATION SCHEMES

First, we set $K = 65,751$ and use a set of TCRs that are known to be associated with the set of labels we are interested in. Figure 14 provides a clear picture of how JL-Norm initialization compares to full GloVe- training with random initialization of TCR embeddings. We see that JL-Norm embeddings already yield non-trivially good performance metrics, which quickly reaches full-training level performance after fine-tuning on a very small portion of $C$ for only a few epochs. Note that same level of training with random initialization performs very poorly. Next, we increase to $K = 360,596$ to assess how well can our methods scale and stay robust to noise as we include more and possibly unrelated TCRs for the tasks we consider, see Figure 15. We note that fine-tuning JL vs. full training compares similar to before, and the performance decrease compared to $K = 65,751$ is not significant in general. This is critical as we would like to include as many TCRs as possible in our analyses to include more HLA/exposure coverage (see Section 4.2).

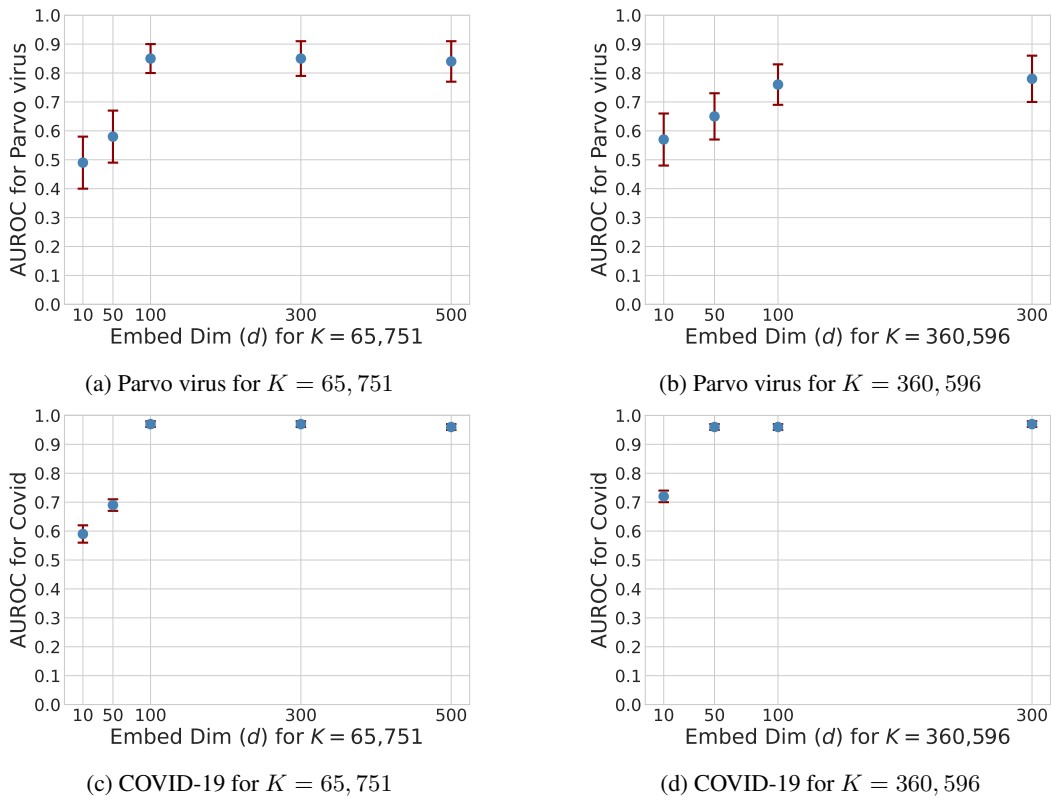

Figure 13: JL-GLOVE AUROC for COVID-19 and Parvo virus across varying numbers of TCRs $K$ and embedding dimensions $d$ on the MULTIID test dataset. We report the median AUROC with a 95% CI from 100 bootstrap samples.

Table 7: Comparison of HSV disease models on MULTIID test data. We report the AUROC, AUPRC, and sensitivity at 98% specificity, both overall and stratified by subtype. We present the median and 95% CI from 100 bootstrap samples for the models AIRIVA and ESLG from Pradier et al. (2023), and JL-GloVe ($d = 100$; $N = 31,938$; varying HLADB TCRs $K$).

| HSV-1 Model | Overall | | | HSV-2 negative | | | HSV-2 positive | | |
|---|---|---|---|---|---|---|---|---|---|
| | Sensitivity | AUROC | AUPRC | Sensitivity | AUROC | AUPRC | Sensitivity | AUROC | AUPRC |
| ESLG (Pradier et al., 2023) | $0.12 \pm 0.10$ | $0.62 \pm 0.09$ | - | $0.18 \pm 0.15$ | $0.63 \pm 0.12$ | - | $0.14 \pm 0.17$ | $0.50 \pm 0.19$ | - |
| AIRIVA (Pradier et al., 2023) | $0.30 \pm 0.12$ | $0.74 \pm 0.09$ | - | $0.35 \pm 0.20$ | $0.74 \pm 0.10$ | - | $0.32 \pm 0.22$ | $0.67 \pm 0.16$ | - |
| JL-GloVe ($K = 65,751$) | $051 \pm 0.11$ | $0.90 \pm 0.04$ | $0.96 \pm 0.01$ | $0.58 \pm 0.16$ | $0.92 \pm 0.05$ | $0.97 \pm 0.01$ | $0.37 \pm 0.17$ | $0.78 \pm 0.12$ | $0.92 \pm 0.03$ |
| JL-GloVe ($K = 360,596$) | $0.46 \pm 0.15$ | $0.87 \pm 0.05$ | $0.94 \pm 0.01$ | $0.57 \pm 0.14$ | $0.91 \pm 0.05$ | $0.96 \pm 0.01$ | $0.40 \pm 0.22$ | $0.81 \pm 0.11$ | $0.93 \pm 0.03$ |
| JL-GloVe ($K = 3,796,900$) | $0.09 \pm 0.11$ | $0.64 \pm 0.09$ | $0.82 \pm 0.03$ | $0.03 \pm 0.09$ | $0.64 \pm 0.12$ | $0.76 \pm 0.06$ | $0.27 \pm 0.19$ | $0.65 \pm 0.14$ | $0.88 \pm 0.05$ |

(a) HSV-1 Prediction Task

| HSV-2 Model | Overall | | | HSV-1 negative | | | HSV-1 positive | | |
|---|---|---|---|---|---|---|---|---|---|
| | Sensitivity | AUROC | AUPRC | Sensitivity | AUROC | AUPRC | Sensitivity | AUROC | AUPRC |
| ESLG (Pradier et al., 2023) | $0.11 \pm 0.10$ | $0.75 \pm 0.07$ | - | $0.16 \pm 0.21$ | $0.79 \pm 0.16$ | - | $0.12 \pm 0.15$ | $0.75 \pm 0.10$ | - |
| AIRIVA (Pradier et al., 2023) | $0.37 \pm 0.18$ | $0.78 \pm 0.10$ | - | $0.57 \pm 0.26$ | $0.86 \pm 0.12$ | - | $0.32 \pm 0.20$ | $0.77 \pm 0.10$ | - |
| JL-GloVe ($K = 65,751$) | $0.37 \pm 0.22$ | $0.90 \pm 0.04$ | $0.84 \pm 0.04$ | $0.21 \pm 0.73$ | $0.98 \pm 0.04$ | $0.91 \pm 0.09$ | $0.30 \pm 0.18$ | $0.87 \pm 0.06$ | $0.80 \pm 0.05$ |
| JL-GloVe ($K = 360,596$) | $0.25 \pm 0.16$ | $0.86 \pm 0.05$ | $0.78 \pm 0.05$ | $0.21 \pm 0.71$ | $0.97 \pm 0.05$ | $0.91 \pm 0.09$ | $0.23 \pm 0.13$ | $0.82 \pm 0.07$ | $0.73 \pm 0.06$ |
| JL-GloVe ($K = 3,796,900$) | $0.08 \pm 0.10$ | $0.72 \pm 0.07$ | $0.55 \pm 0.06$ | $0.07 \pm 0.22$ | $0.77 \pm 0.12$ | $0.54 \pm 0.11$ | $0.12 \pm 0.14$ | $0.70 \pm 0.09$ | $0.56 \pm 0.06$ |

(b) HSV-2 Prediction Task

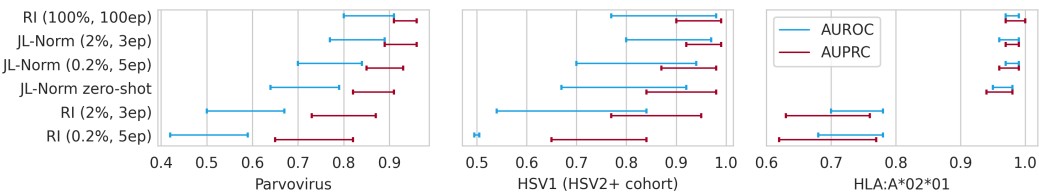

Figure 14: $K = 65,751$ TCRs used. Area under receiver operator characteristics (AUROC) and precision-recall curves (AUPRC).

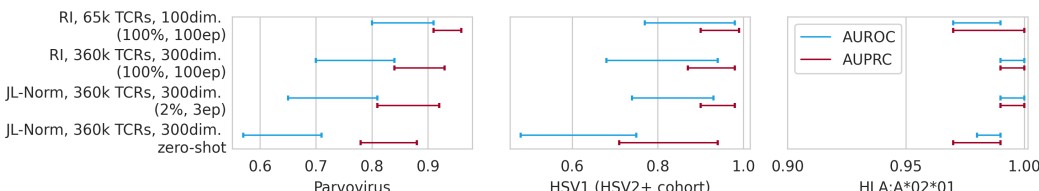

Figure 15: $K = 360,596$ TCRs used. Area under receiver operator characteristics (AUROC) and precision-recall curves (AUPRC).

