# OpenReview forum: "Scalable Universal T-Cell Receptor Embeddings from Adaptive Immune Repertoires"
_ICLR.cc/2025/Conference — ICLR 2025 Poster_

### Official Review · Reviewer_DdmS · 2024-11-03

**Soundness:** 3
**Presentation:** 3
**Contribution:** 3
**Rating:** 6
**Confidence:** 4

**Summary:**

This paper introduces a scalable approach to generating T-cell receptor (TCR) embeddings by leveraging the GloVe algorithm, adapted with the Johnson-Lindenstrauss (JL) transform for improved computational efficiency. The approach aims to create subject-level embeddings of TCR repertoires, which capture immune genetics and pathogenic exposure history. It employs a co-occurrence-based model to detect immune-related patterns and provides an aggregation of TCR embeddings at the subject level, which the authors claim could serve in predicting diseases and HLA types.

**Strengths:**

- The approach is somewhat novel in applying co-occurrence modeling, inspired by NLP, to TCR data. Leveraging random projection (JL transform) to enhance GloVe's performance also demonstrates creativity in handling large datasets.
- The paper is well-organized with clear methodological sections, providing figures and tables to explain the model architecture and performance comparisons.

**Weaknesses:**

- While the method adapts the GloVe and JL transform for TCR analysis, there is limited advancement in the biological interpretability of embeddings over existing approaches.
- The paper lacks rigorous benchmarks against established methods beyond simple logistic regression. Disease and HLA classification tasks do not adequately demonstrate the model’s robustness, especially with limited sensitivity for certain conditions (e.g., HSV) at larger embedding scales.
- While clustering by disease and antigen provides some interpretative insight, the embeddings’ clinical relevance is unclear.
- Despite using the JL transform to improve scalability, the computational requirements for large-scale TCR data (e.g., 4 million TCRs) are still high, limiting the practical applicability of this approach in settings with constrained computational resources.
- The method relies heavily on co-occurrence patterns, which may not fully account for complex immunological interactions, such as those involving low-frequency, yet clinically relevant, TCRs. Moreover, the assumption that TCRs responding to the same antigen will necessarily co-occur in similar contexts lacks validation and may oversimplify TCR functional diversity.

**Questions:**

- How does the model handle rare but potentially significant TCRs? Given the emphasis on co-occurrence, it is unclear how rare TCRs are represented, as these could provide unique insights in immune responses but may not frequently co-occur with other TCRs.
- The current validation relies primarily on logistic regression without exploring other classifiers or model interpretability techniques, which would strengthen the paper’s claims on model generalizability and utility.
- How does the embedding perform across other immunological datasets?
- Additional clarity on computational scaling challenges would be helpful, especially given the potential high-dimensional space of TCR repertoires.
- What additional features could improve biological relevance?

---

> ### Author Response · Authors · 2024-11-19
> **Rebuttal by Authors**
>
> Thanks for the insightful comments. On i) clinical relevance and ii) model interpretability, please refer to the overall response above.
>
> > The paper lacks rigorous benchmarks against established methods beyond simple logistic regression. Disease and HLA classification tasks do not adequately demonstrate the model’s robustness, especially with limited sensitivity for certain conditions (e.g., HSV) at larger embedding scales.
>
> **Response**
>
> The benchmarks used for comparison are the most rigorous available. Prior work found little to no
> improvement over logistic regression for disease prediction tasks (Pradier et al., 2023; Kanduri et al.,
> 2022). *Please refer to the shared response above on the motivation for classification tasks and comparisons
> with prior works for more details*.
>
> The limited sensitivity observed in certain tasks reflects the complex and inherently ambiguous biology
> rather than model limitations. For instance, a challenge with HSV is that there are two closely
> related viruses, HSV-1 and HSV-2, which share a significant portion of their genome, resulting in a
> subset of identical antigens presented to the immune system when infected by either virus. Distinguishing
> whether HSV-specific TCRs recognize unique antigens from HSV-1 or HSV-2, or shared
> antigens presented by both, is inherently difficult for any model given the sparsity and high dimensionality
> of repertoire data. Notably, the *task-agnostic* JL-Glove repertoire embeddings perform as
> well as the AIRIVA *task-specific* model trained to disentangle these two viruses (Table 3).
>
> > How does the model handle rare but potentially significant TCRs? Given the emphasis on co-occurrence, it is unclear how rare TCRs are represented, as these could provide unique insights in immune responses but may not frequently co-occur with other TCRs.
>
> **Response**
>
> Due to their enormous range in generation probability through random V(D)J recombination, a significant
> fraction of TCRs are only observed in one subject and thus are not part of our co-occurrence
> analysis. Our goal however is not to capture all possible signals in the TCR repertoire but to aggregate
> sufficient information to associate a large fraction of embedded TCRs with HLAs and diseases.
> By aggregating these associations, we generate a dense vector representation that encodes both immune
> genetics and disease exposure history. The method presented is a significant step toward this
> objective. As datasets grow and more repertoires are sequenced, we will capture additional signals
> from rarer TCRs (Figure 5).
>
>
> > How does the embedding perform across other immunological datasets?
>
> **Response**
>
> We focus on learning TCR embeddings from bulk TCR sequencing data, as it provides deep sampling, each repertoire is characterized by TCRs in $O(10^7)$ and a large sample size $N$, which is crucial for learning meaningful TCR embeddings, as illustrated in Figures 5 and 10. Most publicly available labeled single-cell TCR sequencing data have low sample sizes in the order $< 30$ samples, and the number of TCRs per patient is even lower $O(10^4)$, making it difficult to benchmark on repertoire classification tasks. However, the proposed TCR embeddings could be combined with other immunological representations, both at the T cell level (e.g., single-cell RNA) and the patient level (e.g., B cell receptors repertoires and histopathology), to form a comprehensive mulit-modal representation.
>
> > Additional clarity on computational scaling challenges would be helpful, especially given the potential high-dimensional space of TCR repertoires
>
> **Response**
>
> JL-GloVe addresses scalability in training the GloVe objective in two ways: (i) computing the normalized JL embeddings for a large number of TCRs $K$ is extremely fast since we don't need to compute the co-occurrence matrix $C$ (see Equations 2, Proposition 2.2, and Algorithm 2 in the Appendix), and (ii) by initializing with normalized JL embeddings, the GloVe algorithm converges faster in fewer epochs, enabling training with only a fraction of the co-occurrence matrix $C$ while achieving similar performance on downstream tasks (see Figures 3, 14, and 15). Additionally, we compute the co-occurrence matrix on CPUs using distributed learning in Spark according to Algorithm 1 in the Appendix and leverage PyTorch Lightning's *distributed data parallel training* to train across GPUs (see Appendix A.6). We understand the resource constraints within the community, therefore, all TCR embeddings derived from the public cohorts, along with the PyTorch code, will be made publicly available.
>
> > What additional features could improve biological relevance?
>
> **Response**
>
> The experimental results show that the proposed TCR embeddings encode immune genetics
> (HLA) and pathogenic exposure history. Additional multi-modal learning with other T cell modalities,
> such as single-cell RNA, could enhance the encoding of TCR function. We consider this
> important future work.

---

> > ### Author Response · Authors · 2024-11-25
> > **Rebuttal Response by Authors**
> >
> > Dear Reviewer,
> >
> > Thank you for your review and insightful comments. We would like to confirm whether you have read the rebuttal and to know if there are any concerns we have not addressed.
> >
> > We kindly ask if you would consider raising your score if most of your concerns have been addressed.
> >
> > Please note, the deadline to interact with the authors is tomorrow.

---

> > ### Comment · Reviewer_DdmS · 2024-11-27
> >
> > Most of my previous comments have been addressed.

---

> > > ### Comment · Reviewer_DdmS · 2024-11-27
> > >
> > > After revision, I think the score can be increased to 6-7

---

> ### Author Response · Authors · 2024-11-27
> **Response by Authors**
>
> We thank the reviewer for engaging during the discussion period. We are deeply grateful that they found our rebuttal addressed most of their concerns and promised to increase their score to recommend acceptance of our paper upon revision.
>
> We would like to inform the reviewer that a revised version of our paper, with changes highlighted in blue, has been uploaded. Additionally, we gently remind the reviewer that they can currently adjust their score during the discussion period.

---

> > ### Author Response · Authors · 2024-12-02
> > **Gentle Reminder by Authors**
> >
> > Dear Reviewer,
> >
> > We are deeply grateful that our rebuttal addressed most of your concerns. A revised version of our paper, with changes highlighted in blue, has been uploaded.
> >
> > We gently remind you that you may adjust your score during the discussion period by clicking the **Edit** button next to your official review.

---

### Official Review · Reviewer_x8Nf · 2024-11-04

**Soundness:** 4
**Presentation:** 2
**Contribution:** 3
**Rating:** 8
**Confidence:** 4

**Summary:**

This paper develops JL-GLOVE, a method for creating vector representations/embeddings of T-cell receptors (TCRs) and immune repertoires that capture meaningful biological relationships. The method leverages TCR co-occurrence across patient repertoires, adapting the GloVe algorithm from natural language processing while incorporating the Johnson-Lindenstrauss transform for computational efficiency. TCRs are embedded such that those targeting the same pathogen have similar vector representations, and patient repertoires are represented by averaging their constituent TCR embeddings. The resulting embeddings successfully encode both immune genetics (HLA types) and pathogen exposure history, improving as more data is added, and outperform baseline methods on disease prediction tasks. The authors demonstrate their method's scalability and interpretability, showing it can process millions of TCRs while maintaining performance, though they note that the simple averaging approach for patient-level representations could be improved. By creating these biologically meaningful representations, the work provides a foundation for quantifying immune system similarity between individuals and could assist in personalized medicine applications.

**Strengths:**

- Originality: the development of TCR embeddings and immune repertoire representations is original and an under studied area in the representation learning community. The application of Glove algorithm here fits nicely and works.
- Quality: the produced results are of high quality and provide a significant impact to the field
- Clarity: the paper is very clear to read and understand. The authors give the right amount of biological/immunological background to understand the paper and why it is important.
- Significance: this is a very significant and meaningful contribution to the field of personalized medicine. The application of representation learning for TCRs and immune repertoires is a great step towards better medicine.

**Weaknesses:**

- The novelties of the paper are not the representation learning method itself. The paper applies Glove algorithm with a few modifications that help it work better, but are not necessarily innovations in of themselves.
- As stated, the immune repertoire method of taking the average, is nice and works, it could be further developed by other methods.

**Questions:**

- I believe this paper should be in the topic area of applications to physical sciences (biology / immunology) rather than unsupervised, self-supervised, semi-supervised, and supervised representation learning. There are few novelties in terms of methods development, but the application of these methods are extremely impactful and a nice way to show the power of representation learning.
- You mention other methods for set level representations, why not use those? Average is nice in it's simplicity, but does this simplicity cost performance? Would be nice to see a benchmark against set representation methods. You may also be interested in OTKE method for set level representation.

---

> ### Author Response · Authors · 2024-11-19
> **Rebuttal by Authors**
>
> Thanks for the insightful comments.
>
> > I believe this paper should be in the topic area of applications to physical sciences (biology / immunology) rather than unsupervised, self-supervised, semi-supervised, and supervised representation learning. There are few novelties in terms of methods development, but the application of these methods are extremely impactful and a nice way to show the power of representation learning.
>
> **Response**
>
> We agree that the application of the JL-GloVe algorithm to TCR embeddings is a novel contribution
> and fits well within the immunology community. However, this work will also be of interest
> to the broader machine learning community for two key reasons: i) The proposed JL-GloVe algorithm
> is general and could be repurposed to learn scalable co-occurrence-based representations with
> lower memory usage and faster training time, especially in contexts where data structures do not
> align with current representation learning paradigms used in other domains such as text or images;
> and ii) Multi-modal representation learning at both the T cell and patient levels will benefit from
> TCR embeddings that account for immune genetics and exposure history. Given these aspects, we
> prefer to maintain our current topic area.
>
> > You mention other methods for set level representations, why not use those? Average is nice in it's simplicity, but does this simplicity cost performance? Would be nice to see a benchmark against set representation methods. You may also be interested in OTKE method for set level representation.
>
> **Response**
>
> Thank you for bringing the OTKE method to our attention; we have included it as an alternative
> set pooling method in the paper (Section 3). Refer to shared response above on *generating TCR repertoire embeddings* for more details.

---

> > ### Comment · Reviewer_x8Nf · 2024-11-22
> > **Rebuttal Response by Reviewer**
> >
> > Hello and thanks for your rebuttal. You've made a fair point. We will keep our rating the same at 6, as we still believe that a larger novelty for methodology paper could have been made. Thanks.

---

> > > ### Author Response · Authors · 2024-11-22
> > > **Rebuttal Response by Authors**
> > >
> > > We thank the reviewer for engaging during the discussion period and for maintaining their positive evaluation score.

---

> > > > ### Comment · Reviewer_x8Nf · 2024-11-27
> > > > **Score Increase**
> > > >
> > > > After a final, and thorough evaluation of your paper, I have decided to increase the score to 8. This is clearly an acceptable paper for ICLR. Thanks.

---

> > > > > ### Author Response · Authors · 2024-11-27
> > > > > **Response by Authors**
> > > > >
> > > > > We are deeply grateful to the reviewer for thoroughly evaluating our paper, increasing their score, and recommending its acceptance at ICLR.

---

### Official Review · Reviewer_ovF4 · 2024-11-04

**Soundness:** 3
**Presentation:** 3
**Contribution:** 4
**Rating:** 8
**Confidence:** 2

**Summary:**

The authors present JL-GLOVE, a scalable algorithm for generating low-dimensional embeddings for T cell receptors (TCRs) and TCR repertoires using TCR co-occurrence data. The main idea is to leverage the co-occurrence patterns of TCRs that target the same antigen to learn meaningful representations. To address the computational challenges of large-scale TCR data, the authors introduce the JL-GLOVE method, which combines GloVe with random projection theory. This approach improves memory efficiency and speeds up the training process. They then aggregate these TCR embeddings to generate subject-level embeddings, providing a low-dimensional representation of an individual's immune history. The embeddings show that TCRs targeting the same antigen exhibit high cosine similarity, and aggregated repertoire embeddings correlate with immune profiles, supporting disease prediction and HLA inference tasks. Results demonstrate the utility of these embeddings for predictive modeling and potential applications in personalized medicine by integrating them with other data modalities.

**Strengths:**

- The use of the JL transform significantly improves the computational efficiency of the GloVe algorithm, enabling the analysis of large datasets containing millions of TCRs. The authors demonstrate that JL-GLOVE achieves good performance using only a fraction of the co-occurrence data, making it suitable for handling the increasing scale of TCR repertoire sequencing data.

- The embeddings produced not only capture the co-occurrence patterns among TCRs but also demonstrate clustering by antigen specificity and HLA association. This biologically meaningful structure aligns with immune response patterns and enhances the interpretability of the embeddings, which is valuable for immunological research and practical applications like personalized medicine.

- The paper rigorously validates the embeddings’ effectiveness through multiple downstream tasks, including disease classification and HLA inference. The experiments demonstrate the robustness of the embeddings to scale, supporting their utility in predicting immune response profiles across various pathogens, and showcasing meaningful performance improvements with larger datasets.

**Weaknesses:**

- The authors compare JL-GLOVE to protein sequence-based embeddings (e.g., ESM-2 and TCRdist), which are structurally different from co-occurrence embeddings. While this comparison is useful, the paper could benefit from a broader comparison with other immunology-focused embedding techniques, such as contrastive learning methods or graph-based embeddings, which may capture additional biological context.

- The paper relies primarily on a mean pooling approach for aggregating TCR embeddings at the repertoire level, which, while straightforward, may be overly simplistic. This method is prone to noise, especially as the number of TCRs (K) increases, potentially limiting classification performance for diseases with more subtle immune signatures.

**Questions:**

- The paper benchmarks JL-GLOVE against ESLG and AIRIVA for disease classification tasks. The authors can include a more comprehensive comparison with other deep learning models specifically designed for TCR repertoire analysis (DeepTCR, DeepID etc).

- The authors observe that the disease classification performance is sensitive to the embedding dimension (d) and the number of TCRs (K). A more systematic exploration of the impact of these parameters can be done A more detailed analysis of the impact of different embedding dimensions across various dataset sizes would be valuable.This would aid other researchers in configuring JL-GLOVE for datasets of different sizes or resolutions, thereby increasing the framework’s accessibility and practical utility.

- Presenting one or two practical case studies where JL-GLOVE embeddings provide actionable insights in a real-world immunological context (e.g., identifying rare disease signatures) would further emphasize the method’s applicability.

---

> ### Author Response · Authors · 2024-11-19
> **Rebuttal by Authors**
>
> Thanks for the insightful comments. On i) comparisons to previous works, ii) generating TCR repertoire embeddings and iii) clinical relevance, please refer to the overall comments above.
>
> >  The authors observe that the disease classification performance is sensitive to the embedding dimension (d) and the number of TCRs (K). A more systematic exploration of the impact of these parameters can be done A more detailed analysis of the impact of different embedding dimensions across various dataset sizes would be valuable.This would aid other researchers in configuring JL-GLOVE for datasets of different sizes or resolutions, thereby increasing the framework’s accessibility and practical utility.
>
> **Response**
>
> We seek to highlight Figure 13, which provides additional experimental results exploring the sensitivity of the dimension $d$ and the number of TCRs $K$, indicating that in general, we need embedding dimensions $d \ge 100$ to obtain reasonable disease prediction performance. Further, theoretical results (Theorem 2.1) show that $d=O(\epsilon^{-2} \log K)$ for distortion $0<\epsilon^{-2}<1$.

---

> > ### Author Response · Authors · 2024-11-25
> > **Rebuttal Response by Authors**
> >
> > Dear Reviewer,
> >
> > Thank you for your review and insightful comments. We would like to confirm whether you have read the rebuttal and to know if there are any concerns we have not addressed.
> >
> > Please note, the deadline to interact with the authors is tomorrow.

---

> > > ### Comment · Reviewer_ovF4 · 2024-11-26
> > >
> > > I would like to thank the authors for their response. I would like to maintain my previous score of 8.

---

> > > > ### Author Response · Authors · 2024-11-26
> > > > **Rebuttal Response by Authors**
> > > >
> > > > We thank the reviewer for engaging during the discussion period and for maintaining their positive evaluation score.

---

### Official Review · Reviewer_Qp7p · 2024-11-04

**Soundness:** 3
**Presentation:** 2
**Contribution:** 3
**Rating:** 5
**Confidence:** 5

**Summary:**

In this study, the authors developed a method to derive low-dimensional representations of T cell receptors and subject-level repertoires in feature space. To enhance computational efficiency, the method employs random projection theory.

**Strengths:**

The paper is well-organized and clearly written.

The proposed method is technically sound.

The application of random projection theory to enhance computational efficiency, particularly regarding memory usage and training time, is noteworthy.

**Weaknesses:**

The biological definitions presented in the study are somewhat unclear. For instance, when the authors refer to TCR embedding, it is important to specify whether they mean both the TCR alpha and beta full chains, the CDR3 regions of both chains, or only the CDR3 region of the TCR beta chain. Additionally, do the authors take into account V(D)J gene information when using the CDR3 region?

Given that TCRs are highly cross-reactive, the authors need to provide further explanation on why using co-occurrence information alone is effective for TCR embedding.

The repertoires of different subjects contain varying numbers of TCRs. How do the authors address this variability when representing them with a matrix of the same TCR dimensionality?

Considering the high cross-reactivity of TCRs, how do the authors define the TCR-level ground truth without relying on wet-lab-based experiments?

When discussing classification tasks, it would be helpful to clarify whether the focus is on receptor-level classification or repertoire-level classification. Furthermore, given different receptors have clone frequencies within the repertoire, it appears that the authors do not consider clone frequency in their repertoire-level embedding.

The interpretation of deep learning models is crucial for clinical applications; however, the authors have provided limited results in this area.

Lastly, there is a noticeable lack of comprehensive comparisons with state-of-the-art works such as DeepTCR, TCRAI, DeepAIR, and DeepRC, which should be addressed.

**Questions:**

In the weaknesses section, it would be beneficial to provide further illustrations regarding the biological background, the methodology employed, and a detailed explanation of why the model is effective.

Moreover, in addition to reporting AUC and sensitivity, the authors should also include other relevant metrics such as specificity, positive predictive value (PPV), negative predictive value (NPV), and overall accuracy. It is important for the authors to clarify how the cut-off points for these metrics were determined, as this information is crucial for understanding the model's performance and its clinical applicability.

---

> ### Author Response · Authors · 2024-11-19
> **Rebuttal by Authors**
>
> Thanks for the insightful comments. On  i) interpretability and ii) comparisons to previous works, please refer to the overall comments above.
>
> > The biological definitions presented in the study are somewhat unclear. For instance, when the authors refer to TCR embedding, it is important to specify whether they mean both the TCR alpha and beta full chains, the CDR3 regions of both chains, or only the CDR3 region of the TCR beta chain. Additionally, do the authors take into account V(D)J gene information when using the CDR3 region?
>
> **Response**
>
> We thank the reviewer for this comment and have clarified in Section 2 that we embed the bioidentity of the TCR$\beta$ chain, where the bioidentity is defined as the CDR3+Vgene+Jgene. However, the JL-GloVe algorithm is general and could be repurposed to learn embeddings of the TCR$\alpha$ chain or both chains, given large TCR repertoire datasets.
>
> > Given that TCRs are highly cross-reactive, the authors need to provide further explanation on why using co-occurrence information alone is effective for TCR embedding.
>
> **Response**
>
> TCRs do indeed have a high theoretical potential for cross-reactivity. However, evidence suggests
> that meaningful cross-reactivity in vivo is rare (Ishizuka et al., 2009; Petrova et al., 2012).
> Previous work using disease-labeled repertoire datasets have also identified extensive sets of TCRs
> strongly associated with individual diseases (Emerson et al., 2017; Greissl et al., 2021; May et al.,
> 2024) and thus not cross-reactive to any common antigenic exposures. One exception are homologous
> antigens, like those responding to HSV-1 and HSV-2, or other homologous viruses.
>
> > The repertoires of different subjects contain varying numbers of TCRs. How do the authors address this variability when representing them with a matrix of the same TCR dimensionality?
>
> **Response**
>
> In Section 3, we detail how we derive repertoire embeddings, given a varying number of TCRs
> per repertoire (subject), via a set mean pooling transformation of TCR embedding dimensions, as
> specified in Equation 8. Refer to overall comment above on *generating TCR repertoire embeddings* for more details.
>
> > Considering the high cross-reactivity of TCRs, how do the authors define the TCR-level ground truth without relying on wet-lab-based experiments?
>
> **Response**
>
> Previous work has identified TCRs associated with infections such as CMV (Emerson et al., 2017),
> SARS-CoV-2 (Snyder et al., 2020), EBV, HSV-1, HSV-2, Parvovirus (May et al., 2024), and Lyme
> disease (Greissl et al., 2021), achieving high sensitivity and specificity in distinguishing cases from
> controls. The observed high specificity of these TCRs, which make up a substantial fraction of
> our embedded TCRs, would not be achievable if substantial cross-reactivity were prevalent in vivo.
> In the context of SARS-CoV-2 infections, specificity of TCRs *(i.e., ground level truth)* has been
> established via wet-lab experiments using functional assays directly demonstrating T cell activation
> by antigens (Snyder et al., 2020).
>
> These in vitro experiments have been validated statistically in vivo. (Pradier et al., 2023) show that
> TCRs purported to bind spike and non-spike derived antigens by MIRA do indeed separate subjects
> who have natural infection from vaccination, further demonstrating that the functional assay used to
> associate TCRs to antigens is reproducing in vivo biology. In this work, we rely on these wet lab
> experiments to provide ground truth for our validation analysis demonstrating that TCRs binding
> antigens derived from the same disease have high cosine similarity (Figure 6a) and to show that subjects who
> have natural infection have greater overlap with TCRs associated with our covid classifier than
> subjects who only have vaccination (Figure 6b).
>
>
> > When discussing classification tasks, it would be helpful to clarify whether the focus is on receptor-level classification or repertoire-level classification.
>
> **Response**
>
> Thank you for the suggestion. We have clarified that all quantitative classification tasks are at
> the repertoire level (Section 4.2) and that all TCR-based results are qualitative. Refer to the shared
> response on *motivation for classification benchmarking tasks* for more details.
>
> > Furthermore, given different receptors have clone frequencies within the repertoire, it appears that the authors do not consider clone frequency in their repertoire-level embedding.
>
> **Response**
>
> We thank the reviewer for this comment and have added text in Section 2 clarifying that we do not
> consider clone frequencies within the repertoire, as we have found binary indicators of the presence
> or absence of TCRs to be more robust.

---

> > ### Author Response · Authors · 2024-11-19
> > **Rebuttal by Authors (Cont.)**
> >
> > > Moreover, in addition to reporting AUC and sensitivity, the authors should also include other relevant metrics such as specificity, positive predictive value (PPV), negative predictive value (NPV), and overall accuracy. It is important for the authors to clarify how the cut-off points for these metrics were determined, as this information is crucial for understanding the model's performance and its clinical applicability.
> >
> > **Response**
> >
> > Thank you for the suggestion. We wish to clarify that all sensitivity results are reported at
> > 98% specificity, as described in Tables 2-7. We agree that a more comprehensive evaluation should
> > include the area under the precision-recall curves (AUPRC) to account for class imbalance, which
> > we provide for K = 65, 751 and K = 360, 596 in Figures 14-15. Complete results, including
> > AUPRC, are now provided in Appendix A.10 (Tables 4-7). We leave the selection of specific cut-off
> > points to the user, as these may differ depending on the application.

---

> > > ### Author Response · Authors · 2024-11-25
> > > **Rebuttal Response by Authors**
> > >
> > > Dear Reviewer,
> > >
> > > Thank you for your review and insightful comments. We would like to confirm whether you have read the rebuttal and to know if there are any concerns we have not addressed.
> > >
> > > We kindly ask if you would consider raising your score if most of your concerns have been addressed.
> > >
> > > Please note, the deadline to interact with the authors is tomorrow.

---

> > > > ### Author Response · Authors · 2024-12-02
> > > > **Gentle Reminder by Authors**
> > > >
> > > > Dear Reviewer,
> > > >
> > > > Thank you for your review and insightful comments. We would like to confirm whether you have read the rebuttal and to know if there are any concerns we have not addressed.
> > > >
> > > > We kindly ask if you would consider raising your score if most of your concerns have been addressed.
> > > >
> > > > Please note, the deadline to interact with the authors is today.

---

### Author Response · Authors · 2024-11-19
**Overall Rebuttal by Authors**

We thank the reviewers for their encouraging and insightful comments. We are grateful for the
largely positive reception of our work and would like to highlight a few key points made by the
reviewers:
* Originality: Leveraging random projection (JL transform) to enhance GloVe’s performance
is a creative and noteworthy approach. *(Qp7p, ovF4, x8Nf, DdmS)*
* Clarity: The paper is well-organized, well-written, and easy to understand, with clear
methodological sections. *(Qp7p, x8Nf, DdmS)*
* Soundness: The proposed method is technically sound. Extensive experimental results
demonstrate the robustness of the embeddings at scale and their effectiveness on down-
stream tasks. *(Qp7p, ovF4, x8Nf)*
* Significance: The embeddings capture co-occurrence patterns among TCRs and show clustering by antigen specificity and HLA association, making a significant contribution to personalized medicine. *(ovF4, x8Nf)*

We are encouraged by the positive evaluation scores (8, 6) and are keen to address the concerns
underlying the lower scores (5, 5) to bridge gaps in understanding. Below are the point-by-point
responses and a summary of changes, which are also highlighted in the updated revision.

---

> ### Author Response · Authors · 2024-11-19
> **Overall Rebuttal by Authors (Cont.)**
>
> **Motivation for classification benchmarking tasks (Reviewers Qp7p, DdmS)**
>
> We seek to clarify that the JL-GloVe TCR embeddings encode immune genetics (HLA) and pathogenic exposure history. In Section 4.2, we present quantitative *repertoire-level* classification tasks based on *shared task-agnostic repertoire embeddings* generated according to Equation 8, primarily to validate this hypothesis. Here, by *task-agnostic*, we simply mean that we have not derived these embeddings by optimizing for any specific downstream modeling application. We emphasize our goal is not to build the most optimal model for any specific task but rather to demonstrate the generalizability and robustness of the inferred TCR embeddings across *5 disease prediction and 145 HLA inference* binary classification tasks. We have clarified this in Section 4.2.
>
> **Comparisons to previous works (Reviewers Qp7p, ovF4, DdmS)**
>
> Thank you for bringing these works to our attention. We have included them in the introduction and provided comparisons with DeepRC on the Emerson dataset (Tables 2 and 4). Note that DeepRC reports a best AUROC of $0.831 \pm 0.002$ compared to JL-GloVe's $0.99 \pm 0.02$ and ESLG's $0.93 \pm 0.01$. These results align with previous findings showing that TCR protein sequence models struggle to outperform simple regularized logistic regression models, such as ESLG (Kanduri et al., 2022). We wish to emphasize that the proposed JL-GloVe TCR embeddings are *task-agnostic* and serve as a complement to the TCR protein sequences used as features in supervised learning approaches, including DeepTCR, TCRAI, DeepAIR, and DeepRC. Since JL-GloVe TCR embeddings encode immune genetics (HLA) and pathogenic exposure history, they can be effectively utilized in conjunction with TCR protein sequences for tasks such as repertoire classification or TCRpMHC binding prediction, which are employed in these methods.
> Furthermore, qualitative results (Figure 7) demonstrate that TCR protein sequence-based embeddings fail to cluster the embedding space by HLA or disease exposure, unlike the proposed JL-GloVe co-occurrence-based TCR embeddings. Additionally, the JL-GloVe algorithm is general and could be repurposed for other immunological datasets, including B-cell receptor repertoires used in DeepID. We have emphasized this aspect in Sect 2.3
>
> **Clinical relevance (Reviewers ovF4, DdmS)**
>
> We illustrate the power of JL-GloVe embeddings by showing that they can distinguish
> between individuals vaccinated against COVID-19 and those with natural infections based
> on distinct co-occurrence patterns (Figure 6b). A similar separation is observed for TCRs
> associated with HSV-1 and HSV-2 infections (Table 3). Our approach can facilitate similar
> analyses for other homologous viruses where labels may be unavailable. The
> most important clinical applications will be in the ability to study the interdependence
> between diseases. For instance, a strong link between EBV infection and multiple sclerosis
> has recently been identified by Bjornevik et al. (2022); uncovering this link required disease
> exposure history for a large cohort of patients. We plan to release embeddings trained on
> public cohorts, which we hope will benefit research groups with smaller datasets to explore
> disease interdependency.
>
> **Generating TCR repertoire embeddings (Reviewers Qp7p, ovF4, x8Nf)**:
>
> We opted for *parameter-free* mean pooling primarily for simplicity and easier interpretability
> in repertoire classification tasks, as we wanted to highlight that the TCR embeddings
> capture immune genetics and pathogenic exposure history without learning additional
> parameters. The proposed simple mean pooling approach will serve as a strong baseline for
> more sophisticated set pooling approaches, including OTKE and set transformers. We agree
> that experimental results show that as we increase the number of TCRs K, mean pooling
> results in noisier repertoire embeddings. This necessitates the sophisticated methodological
> development that adapts a set-based pooling mechanism to derive *task-agnostic (universal)*
> repertoire embeddings, which account for the challenges posed by TCR repertoire data,
> which are heterogeneous, high-dimensional, sparse, and mostly unlabeled. We consider this to be a key area for future work.
>
> **Interpretability of TCR embeddings (Reviewers Qp7p, DdmS)**:
>
> Note that the JL-GloVe algorithm is derived from TCR co-occurrence to maximize the cosine similarity of co-occurring
> TCRs, along with an interpretable monotonic function derived from marginal
> occurrence to down-weight commonly occurring TCRs. Wet-lab results demonstrate that
> TCRs targeting the same antigen have high cosine similarity (Figures 4a and 6a). Additionally,
> we assume a simple linear mean pooling function for repertoire embeddings, which enables: i) stratification of repertoires by antigen exposure (Figure 6b), and ii) recovery of disease/HLA-associated TCRs (Figure 12) through simple dot products.

---

### Meta-Review · Area_Chair_pDSW · 2024-12-15

**Metareview:**

The paper is at the intersection of Biology and AI. It develops embedding which will be useful for understanding of how Tcells target various infections. There is consensus that this paper should be of interest to ICLR community.

**Additional Comments On Reviewer Discussion:**

Given the interdisciplinary nature of the paper, several concerns related to clinical relevance, choice of benchmarks, and specific technicalities on how the GLOVE embeddings have been adapted via JL Transform for this problem.
The authors addressed most of them and made appropriate changes.

---

### Decision · Program_Chairs · 2025-01-22

Accept (Poster)